# VCP recruitment to mitochondria causes mitophagy impairment and neurodegeneration in models of Huntington's disease

Xing Guo[1], XiaoYan Sun[1], Di Hu[1], Ya-Juan Wang[2], Hisashi Fujioka[3], Rajan Vyas[1], Sudha Chakrapani[1], Amit Umesh Joshi[4], Yu Luo[5], Daria Mochly-Rosen[4] & Xin Qi[1,6]

Mutant Huntingtin (mtHtt) causes neurodegeneration in Huntington's disease (HD) by evoking defects in the mitochondria, but the underlying mechanisms remains elusive. Our proteomic analysis identifies valosin-containing protein (VCP) as an mtHtt-binding protein on the mitochondria. Here we show that VCP is selectively translocated to the mitochondria, where it is bound to mtHtt in various HD models. Mitochondria-accumulated VCP elicits excessive mitophagy, causing neuronal cell death. Blocking mtHtt/VCP mitochondrial interaction with a peptide, HV-3, abolishes VCP translocation to the mitochondria, corrects excessive mitophagy and reduces cell death in HD mouse- and patient-derived cells and HD transgenic mouse brains. Treatment with HV-3 reduces behavioural and neuropathological phenotypes of HD in both fragment- and full-length mtHtt transgenic mice. Our findings demonstrate a causal role of mtHtt-induced VCP mitochondrial accumulation in HD pathogenesis and suggest that the peptide HV-3 might be a useful tool for developing new therapeutics to treat HD.

[1] Department of Physiology & Biophysics, Case Western Reserve University School of Medicine, Cleveland, Ohio 44106, USA. [2] Center for Proteomics and Bioinformatics, Case Western Reserve University School of Medicine, Cleveland, Ohio 44106, USA. [3] Department of Pharmacology, Case Western Reserve University School of Medicine, Cleveland, Ohio 44106, USA. [4] Department of Chemical and Systems Biology, Stanford University School of Medicine, Stanford, California 94043, USA. [5] Department of Neurosurgery, Case Western Reserve University School of Medicine, Cleveland, Ohio 44106, USA. [6] Center for Mitochondrial Disease, Case Western Reserve University School of Medicine, Cleveland, Ohio 44106, USA. Correspondence and requests for materials should be addressed to X.Q. (email: xxq38@case.edu).

Huntington's disease (HD) is a fatal and inherited neurodegenerative disorder. The disease is caused by an abnormal expansion of a CAG repeat located in exon 1 of the gene encoding the huntingtin protein (Htt), which confers a toxic gain of function to the protein[1]. The most striking neuropathology in HD is the preferential loss of medium spiny neurons in the striatum[2]. Although the genetic defect that causes HD has been identified as mutant huntingtin (mtHtt), a causative pathway from the disease mutation gene to neuronal death remains elusive. Neither a cure nor disease-modifying treatment is currently available.

Evidence suggests that mtHtt causes neurotoxicity by evoking defects in mitochondria, which in turn leads to a bio-energetic failure, HD-linked neuronal dysfunction and cell death[3,4]. Recent studies including ours show that mtHtt triggers mitochondrial fragmentation and associated mitochondrial dysfunction by hyper-activating the primary mitochondrial fission protein, Dynamin-related protein 1 (Drp1)[5–7]. Inhibition of Drp1 by either pharmacological inhibitors or the genetic approach rescued mtHtt-induced mitochondrial and neuronal dysfunction in a variety of HD models[5,7]. Moreover, cyclosporine A (ref. 8), an inhibitor of mitochondrial permeability transition pore opening) and trans-(-)-Viniferin (ref. 9), an activator of mitochondrial sirtuin 3), were all protective in HD models. These findings not only provided further evidence that mitochondrial damage plays a causal role in the pathogenesis of HD, but also demonstrated that blocking mitochondrial injury can reduce neuronal pathology in HD.

Mutant Htt localizes to the mitochondria, where it can either recruit soluble cytosolic proteins or interact with mitochondrial components[7,10]. Because altered binding of Htt with target proteins can significantly contribute to the pathogenesis of HD[11], we recently profiled the proteins that bind to mtHtt on mitochondria and identified valosin-containing protein (VCP) as a high-abundance mtHtt-interacting protein on mitochondria (Supplementary Fig. 1). VCP, also known as p97 in vertebrates and Cdc48 in *S. cerevisiae*, is a class II member of the ATPase associated with diverse cellular activities (AAA) ATPase. VCP is highly conserved from archaebacteria to humans, and is located in different subcellular organelles, including the endoplasmic reticulum (ER), mitochondria and nucleus, where it functions in diverse cellular processes including ER-associated protein degradation, mitochondria-associated degradation, autophagy and DNA repair[12]. While the role of VCP in maintaining ER proteostasis has been studied extensively, the importance of VCP-dependent mitochondrial maintenance under normal and stressed conditions is just emerging. VCP can translocate to mitochondria, where it is required for turnover of mitochondrial outer membrane proteins[13,14] and parkin-dependent mitophagy[13,15]. Overexpression of VCP results in mitochondrial fragmentation and cell death in neurons exposed to mitochondrial toxins, such as rotenone, 6-OHDA[16]. Pathogenic mutations in VCP in yeast[17] and *Drosophila*[15,18] cause mitochondrial depolarization, mitochondrial oxidative stress, reduced ATP production, and mitochondrial aggregations. Mice with VCP mutants display mitochondrial degeneration, enhanced autophagy, motor neuron degeneration and early lethality[19–21]. In humans, mutations of the *VCP* gene cause frontotemporal dementia, amyotrophic lateral sclerosis and muscular and bone degeneration, all of which are manifestations of mitochondrial dysfunction[12,22].

Endogenous VCP co-localizes with the polyglutamine-containing aggregates in patients with HD and Machado–Joseph disease[23–25]. VCP can bind directly to multiple polyglutamine disease proteins, including huntingtin, ataxin-1, ataxin-7 and androgen receptors[26,27]. In a transgenic *Drosophila* model expressing a fragment of the polyQ gene carrying either 79 or 92 CAG repeats, an upregulation of VCP expression was observed before cell death, and overexpression of VCP severely enhanced eye degeneration[23,28]. Thus, VCP might be a cell death effector in polyglutamine-induced neurodegeneration. However, whether and how VCP mediates neuronal pathology in HD and whether manipulation of VCP can modify or stop the neuronal degeneration associated with HD are unknown.

In this study, we report for the first time that VCP is aberrantly translocated to the mitochondria and bound to mtHtt in a variety of HD models. This accumulation of VCP on mitochondria results in excessive mitophagy and subsequent neuronal degeneration. Blocking VCP translocation to mitochondria by a novel peptide HV-3 that interferes with VCP and mtHtt interaction, inhibits VCP-mediated mitophagy impairment, and reduces HD-associated neuropathology and motor deficits in HD transgenic mouse models. Our results suggest that VCP recruitment to mitochondria by mtHtt is a crucial step in the initiation of neuropathology in HD.

## Results

**VCP is recruited to mitochondria by mtHtt in HD.** We used HD mouse striatal HdhQ111 (mutant) and HdhQ7 (wild-type, wt) cells to profile the interactors of mtHtt on the mitochondria (Fig. 1a, Supplementary Fig. 1). HdhQ111 and Q7 cells were immortalized from knock-in mice carrying 111 and 7 CAG repeats, respectively, in the mouse *htt* gene[29]. We isolated mitochondria from these cells, and conducted immunoprecipitation (IP) of mitochondrial fractions with anti-MAB2166 antibody that recognizes both wt and mutant (mt) Htt (Supplementary Fig. 1a). Tandem mass spectrometry analysis following affinity purification identified 9 proteins that putatively bound to mitochondria-associated mtHtt in HdhQ111 but not wt Htt in HdhQ7 cells (Fig. 1a, Supplementary Fig. 1b). Among these proteins, VCP was the leading candidate that bound to mtHtt on the mitochondria of HdhQ111 cells (Fig. 1a, Supplementary Fig. 1c).

Before validating the interaction between VCP and mtHtt, we examined whether VCP is localized on mitochondria in models of HD. Western blot analysis of cellular fractionations revealed that VCP was markedly enriched in the mitochondria of HdhQ111 cells relative to those in HdhQ7 cells (Fig. 1b), while there was no increase in the recruitment of VCP to the ER in HdhQ111 cells compared with that of HdhQ7 cells (Fig. 1b). Reduction of mtHtt levels by treatment of HdhQ111 cells with Htt silencing RNA (siRNA) abolished VCP translocation to mitochondria (Fig. 1c), indicating that mtHtt is required for VCP recruitment to mitochondria. Confocal imaging analysis consistently showed increased localization of VCP on the mitochondria, but not on the ER and endosome of HdhQ111 cells, relative to that in HdhQ7 cells (Fig. 1d, Supplementary Fig. 2a). Immunogold electron microscopy (EM) found more particles of immuno-labelled VCP localized on the surface of mitochondria in HdhQ111 cells than that in HdhQ7 cells (Fig. 1e). A similar enrichment of VCP on mitochondria was observed in mitochondrial fractions isolated from the striata of both R6/2 mice at the age of 9 weeks and YAC128 mice at the age of 6 months (Fig. 1f). To test whether VCP accumulation on mitochondria also exists in human HD, we analysed VCP localization on mitochondria by confocal microscopy in the caudate nucleus of postmortem brains from three HD patients and three normal subjects. We observed greater localization of VCP on mitochondria in HD patient brains than in normal subjects (Fig. 1g). These data collectively demonstrate that VCP is recruited to and accumulated on mitochondria in HD.

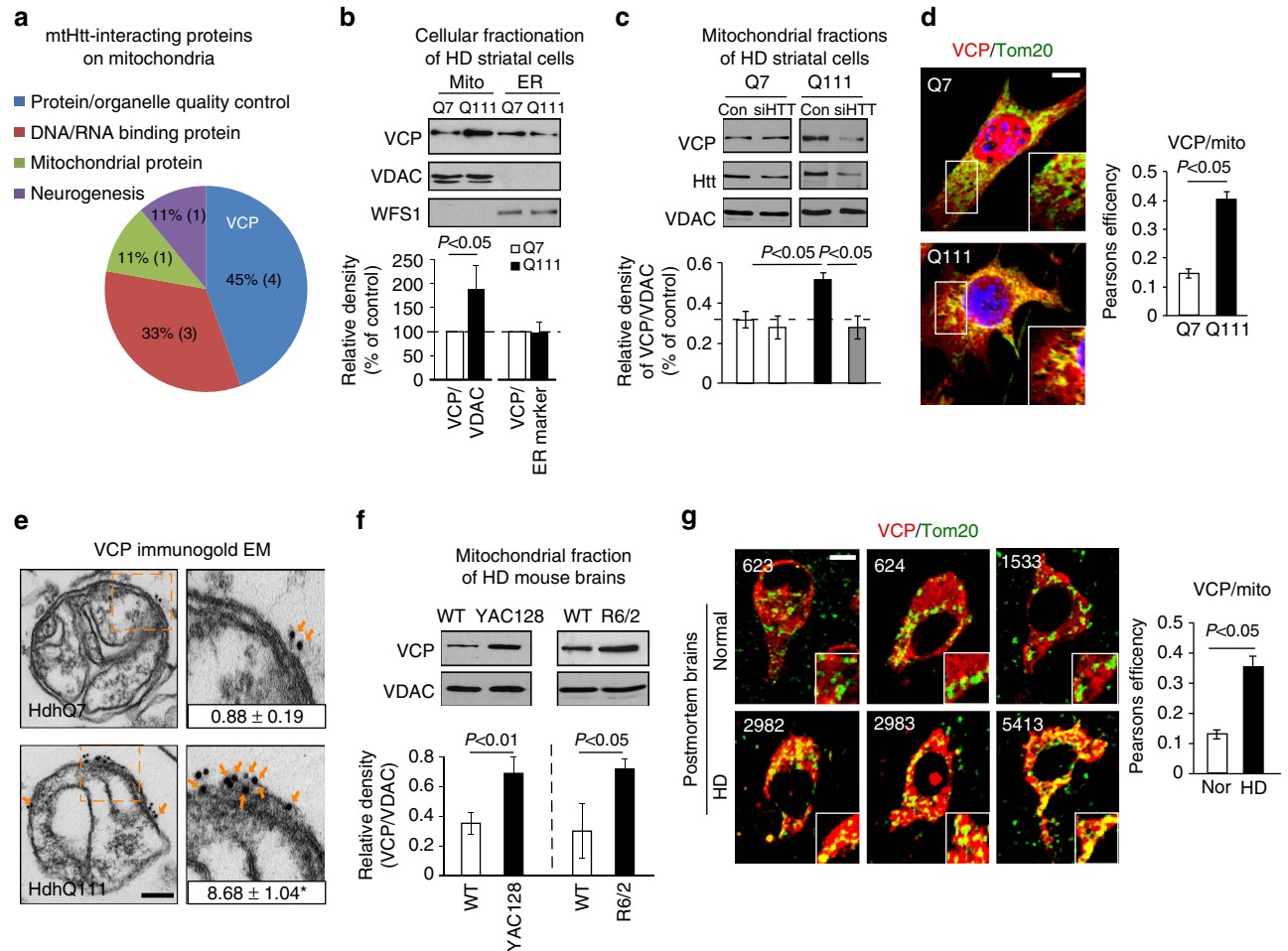

**Figure 1 | VCP is recruited to mitochondria in HD models.** (**a**) Affinity purification followed by tandem mass spectrometry analysis was conducted to identify mtHtt-binding proteins on mitochondria in HdhQ7 and HdhQ111 striatal cells. The molecular and cellular function of the exclusive mtHtt interactors on mitochondria of HdhQ111 cells are shown. VCP was the leading candidate for an mtHtt-binding protein (Supplementary Fig. 1). (**b**) Mitochondrial and ER fractions were isolated from HdhQ7 and HdhQ111 cells. Protein levels of VCP were analysed by western blotting (WB). VDAC and WFS1 were used as loading controls for mitochondria and ER. Data are mean ± s.e.m. of at least three independent experiments. (**c**) Control siRNA (Con) and Htt siRNA (siHTT) were transfected in HdhQ7 and HdhQ111 cells for 3 days, respectively. VCP levels were determined in mitochondrial fractions by WB analysis. VDAC was a loading control. Data are mean ± s.e.m. of at least three independent experiments (One-way ANOVA with Holm-Sidak *post hoc* test). (**d**) HdhQ7 and HdhQ111 cells were stained with anti-Tom20 (green, a mitochondrial marker) and anti-VCP (red) antibodies. VCP/Tom20 co-localization was examined using confocal microscopy. Scale bar: 10 μm. Pearson's co-efficiency was calculated. At least 100 cells per group were counted. Data are mean ± s.e.m. of three independent experiments. (**e**) Immunogold electron microscopy analysis of VCP on mitochondria was conducted. Scale bar: 100 nm. The number of gold particles labelling VCP was quantitated and shown as mean ± s.e.m. A total of 30 mitochondria from each group were counted. *$P < 0.01$ versus HdhQ7 cells. (**f**) Mitochondria were isolated from the striata of either HD transgenic mice R6/2 (9-week-old) or YAC128 (6-month-old). $n = 6$ mice/group. VCP levels were determined by WB (loading control: VDAC). (**g**) Paraffin-embedded sections (5 μm thick) of the caudate nucleus from three HD patients (ID: 2982, 2983 and 5413) and three normal subjects (ID: 623, 624 and 1533) were immunostained with anti-VCP (red) and anti-Tom20 (green) antibodies. Localization of VCP on mitochondria was examined using confocal microscopy. Pearson's co-efficiency was calculated. Scale bar: 10 μm. Data are mean ± s.e.m. (**b,d–g**) Paired Student's *t*-test.

Because there was no evidence of increased VCP recruited to mitochondria in response to Parkinson's disease-associated mutants (Supplementary Fig. 2b), this recruitment is likely to be disease or stress dependent.

**VCP binds to mtHtt on mitochondria in HD.** Next, we isolated mitochondrial, ER and cytosolic fractions from HdhQ7 and HdhQ111 cells, and conducted IP with anti-VCP antibody followed by immunoblotting (IB) with anti-MAB2166 antibody. Surprisingly, we observed mtHtt proteins in VCP immunoprecipitates of mitochondrial fractions, but not in those of ER and cytosolic fractions in HdhQ111 cells (Fig. 2a, left panel).

Although VCP interacted with wt Htt on the mitochondria in HdhQ7 cells, the extent is smaller than in HdhQ111 cells only expressing mtHtt (Fig. 2a, left panel). To further validate the interaction between VCP and mtHtt, we performed IP analysis with anti-VCP antibody followed by IB with anti-1C2 antibody that recognizes only expanded polyQ proteins. As shown in Fig. 2a, right panel, VCP bound only to mtHtt in mitochondrial fractions, not in ER or cytosolic fractions of HdhQ111 cells, even though mtHtt was expressed in the ER and cytosolic fractions. Consistently, mtHtt recognized either by anti-1C2 antibody or by anti-EM48 antibody was observed in VCP immunoprecipitates of mitochondrial fractions isolated from the striata of YAC128 mice at the age of 6 months (Fig. 2b). Again, there was no obvious

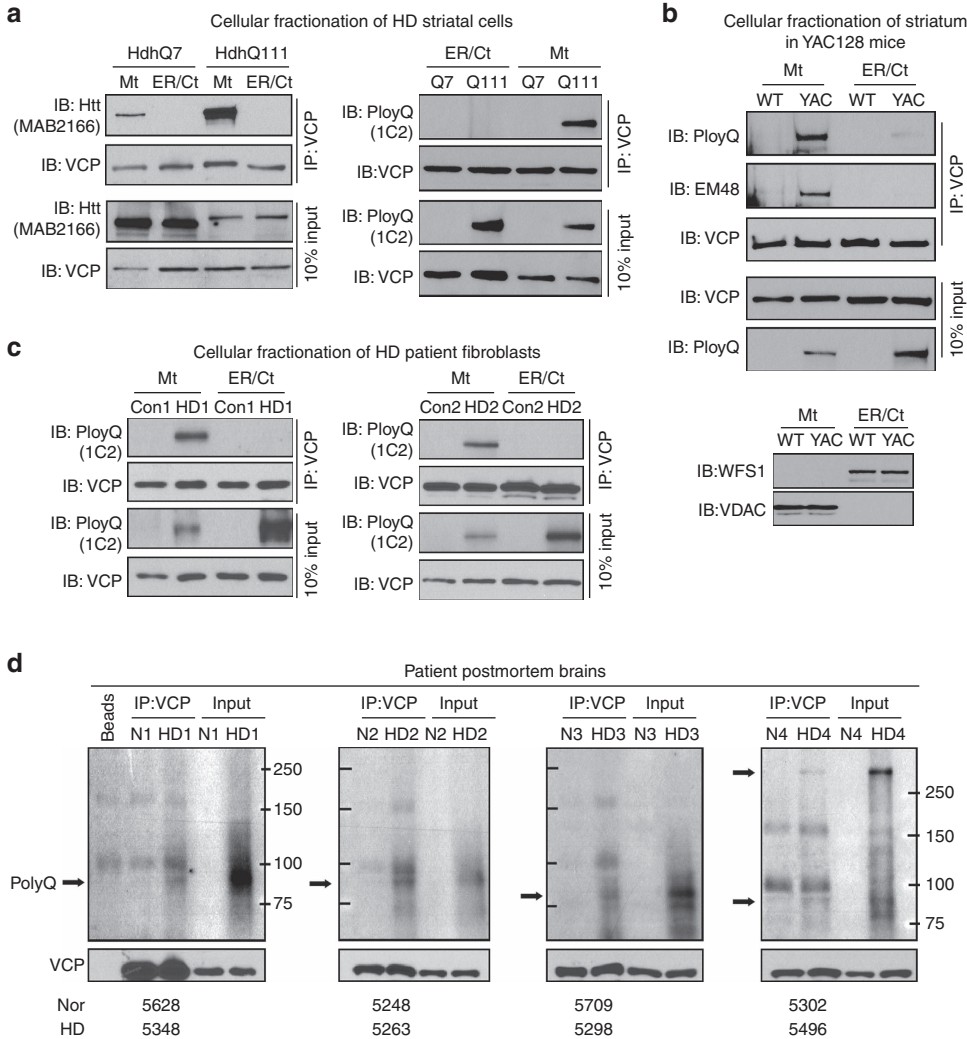

**Figure 2 | VCP binds to mtHtt on mitochondria *in vitro* and *in vivo*.** (**a**) Mitochondrial, ER, and cytosolic fractions (Ct) of HdhQ7 and HdhQ111 mouse striatal cells were subjected to immunoprecipitation (IP) with anti-VCP antibody, and immunoprecipitates were analysed by WB with anti-VCP and anti-MAB2166 antibody (recognizes both wt and mtHtt, left panel) or anti-1C2 antibody (recognizes mtHtt, right panel). Note that polyQ protein above 250 kDa is shown in the right panel. Representative blots from three independent experiments are shown. (**b**) Mitochondrial, ER, and cytosolic fractions were isolated from striata of YAC128 and wild-type mice at the age of 6 months. IP with anti-VCP antibody followed by anti-1C2 antibody or anti-EM48 antibody was performed. The right panel indicates the purity of ER and mitochondrial fractions isolated from YAC128 mouse striata. WFS1 and VDAC were used to label ER and mitochondria, respectively. $n = 4$ mice/group. (**c**) Mitochondrial, ER and cytosolic fractions were isolated from fibroblasts of HD patients (HD1 carries 70 CAG repeats and HD2 carries 60 CAG repeats, respectively) and normal subjects. IP with anti-VCP antibody followed by WB with anti-1C2 antibody was conducted. Representative blots from two independent experiments are shown. (**d**) Total cortical protein lysates from postmortem brain tissues of 4 normal subjects and 4 HD patients were subjected to IP with anti-VCP antibody followed by WB with anti-1C2 antibody. Arrows indicate mtHtt recognized by 1C2 antibody which does not detect wt Htt. Note: mtHtt protein around 82 kDa recognized by 1C2 antibody has been shown to be abundant in cortical tissues of HD mice[68] and HD patients[6]. The identity numbers of the HD patients (HD) and normal subjects (Nor) were listed on the bottom. HD patients (5348 and 5263) exhibited extensive neuronal loss and severe brain atrophy, and HD patients (5298 and 5496) showed moderate neuronal loss and brain atrophy. The information of normal subjects and HD patients was summarized in Supplementary Fig. 2c. Normal subjects had no history of HD or other neurological diseases.

binding of VCP and mtHtt observed in the ER or cytosolic fractions of YAC128 mouse striata (Fig. 2b). We confirmed the interaction of VCP with mtHtt on the mitochondria but not in the ER or cytosolic fractions of HD patient fibroblasts (Fig. 2c). Furthermore, IP analysis of cortical brain lysates from postmortem brain tissues of HD patients showed that VCP bound to mtHtt in HD patients who exhibited moderate to severe neuronal loss and brain atrophy, but not in the patient with subtle neuropathology (Fig. 2d, Supplementary Fig. 2c, d). Altogether, these findings not only support our observation that VCP/mtHtt binding is implicated in HD pathogenesis, but also suggest a relevance of this binding to the severity of HD pathology. A

recent proteomic analysis of the Htt interactome in total brain lysates of BACHD transgenic mice[30] supported our finding that VCP is a binding protein of Htt and that increased interaction between VCP and mtHtt is relevant to HD. Now we are able to locate this interaction with mitochondria in models of HD in culture and in animals, as well as in patient cells. However, the mechanism underlying such a specific interaction between mtHtt and VCP on mitochondria requires further investigation.

VCP plays a central role in protein degradation via the ubiquitin–proteasome system by binding to its substrates[31] and mtHtt can be degraded by the ubiquitin–proteasome system[32]. We found that treatment with either MG132 (a proteasome

inhibitor that prevents protein degradation) or Eeyarestatin I (Eer I, an inhibitor that blocks VCP substrate degradation[33]), did not affect Htt or mtHtt protein levels in HdhQ or HdhQ111 cells, respectively (Supplementary Fig. 3). These data exclude the possibility that Htt or mtHtt is a substrate of VCP on the mitochondria.

**HV-3 peptide interferes with Htt/VCP interaction.** We next ask what specific function VCP mediates on mitochondria from HD models. VCP knock-out is lethal in mice[34]. Compounds that inhibit VCP function, such as N2,N4-dibenzylquinazoline-2,4-diamine and Eeyarestatin I, are non-specific and rapidly lead to cell death[35]. We previously demonstrated that short peptides interfering with specific protein–protein interactions, such as Drp1 peptide inhibitor P110[36] or peptide inhibitors for protein kinase C (PKC, ref. 37), can be used as pharmacological tools in cell, animal, and human models to identify the role of interacting proteins in the pathogenesis of human diseases. Given that mtHtt is required for VCP translocation to mitochondria (Fig. 1c) and that VCP and mtHtt selectively interact on the mitochondria (Fig. 2), we sought to develop a peptide that blocks VCP association with mitochondria by interfering with VCP/Htt interaction.

Similar to the peptide designs for Drp1 peptide P110 or PKC peptide inhibitors[36,37], we used L-ALIGN sequence alignment software[38] and identified two different regions of homology between VCP (human, AAI21795) and Htt (human, NP_002102; Fig. 3a). The four regions are marked as regions HV-1 to HV-4 (Fig. 3b). We synthesized peptides corresponding to the four homologous regions between VCP and Htt (Fig. 3a), and conjugated them to the cell permeating TAT protein-derived peptide, $TAT_{47-57}$, to enable *in vivo* delivery[5,36]. These peptides are referred to as HV-1, HV-2, HV-3 and HV-4. By incubating these peptides with a mixture of GST–VCP and total lysates of mouse brain (expressing full-length Htt) followed by GST pull-down analysis, we found that only the addition of peptide HV-3 blocks the interaction of VCP/Htt in this *in vitro* binding assay (Supplementary Fig. 4a). In HEK293 cells co-expressing Myc-tagged full-length Htt with 23 or 73 CAG repeats (Myc-23Q FL or Myc-73Q FL, respectively) and green fluorescent protein (GFP)-VCP, consistent with our observation (Fig. 2), VCP was preferentially bound to Myc-73Q FL (mtHtt) over Myc-23Q FL (Fig. 3c). Of the four peptides tested, only HV-3 peptide significantly blocked the VCP/mtHtt interaction in Myc-73Q FL expressing cells (Fig. 3c, Supplementary Fig. 4b). The IC50 of HV-3 in blocking VCP/mtHtt interaction in Myc-73Q FL expressing cells was $2.11 \mu M$ (Supplementary Fig. 4c). Peptide HV-3 is derived from the Htt c-terminal and corresponds to a sequence in the D1 domain of VCP (Fig. 3a,b). The sequence of HV-3 in Htt is highly conserved among species (Supplementary Fig. 4d). With the exceptions of Htt and VCP, there is no sequence identity or similarity found between HV-3 and other proteins. Notably, treatment with HV-3 did not influence the interaction of mtHtt with Tim23, an event previously reported[10], nor did it influence the interaction between VCP and its known binding protein UBXD1 (ref. 39; Fig. 3d), suggesting a selectivity of HV-3.

Molecular docking analysis indicates that HV-3 is bound to the surface of the VCP structure (Supplementary Fig. 4e). Deletion of the sequence corresponding to HV-3 in VCP abolished the interaction between Htt and VCP (Supplementary Fig. 4f). Thus, HV-3 may represent an important interaction region for VCP in Htt. To test whether HV-3 exerts its effect through direct interaction with VCP and to determine the affinity of this interaction, we carried out isothermal titration calorimetry (ITC)

with recombinantly expressed and purified full-length VCP (Fig. 3e). The heat exchanged as a result of the interaction between VCP and HV-3 peptide was used to calculate the binding affinity ($K_d$). Our analysis of the binding isotherms clearly showed that HV-3 binds to VCP with a $K_d$ of $17.9 \mu M$ (Fig. 3e). To further examine the specificity of HV-3, we incubated biotin-conjugated HV-3 or TAT with total protein lysates of HD mouse striatal cells and HD YAC128 mouse brain followed by IP analysis. We found that biotin–HV-3, but not biotin–TAT, pulled down VCP and that biotin–HV-3 bound to VCP more strongly in HdhQ111 cells and YAC128 mouse brains relative to that in wt counterparts (Fig. 3f). No detectable bindings were observed between biotin–HV-3 and the cytosolic protein Enolase or between biotin–HV-3 and the mitochondrial protein Clpp (Fig. 3f). Therefore, HV-3 is most likely targeting VCP to interfere with the interaction of VCP/mtHtt.

Next, we test whether the peptide HV-3 influences VCP association with mitochondria in HD models. In HdhQ111 mouse striatal cells, treatment with HV-3 abolished VCP translocation to the mitochondria relative to cells treated with TAT (Fig. 3g). In YAC128 mice, which express a full-length human mtHtt, the treatment blocked VCP translocation to mitochondria in the striatum at the age of 6 months relative to YAC128 mice treated with control peptide TAT (Fig. 3h, treatment timeline in Supplementary Fig. 5a). Similarly, HV-3 treatment suppressed the VCP translocation to mitochondria that occurred in striatum of HD R6/2 mice expressing an N-terminal mtHtt fragment (Fig. 3i, treatment timeline in Supplementary Fig. 5a). HV-3 treatment had no effects on VCP total protein levels in the above HD cell cultures and HD animal brains (Supplementary Fig. 5b). Thus, we selected HV-3 as a peptide candidate to inhibit VCP mitochondrial accumulation and to further determine its activity in HD models.

**HV-3 treatment reduces mitochondrial damage and cell death.** Mitochondrial depolarization and mitochondrial fragmentation are featured in experimental models of HD and human HD[5,40]. Treatment with HV-3 markedly improved the mitochondrial membrane potential (MMP) in HdhQ111 cells, compared with the cells treated with control peptide TAT (Fig. 4a). Downregulation of VCP by VCP siRNA in HdhQ111 cells similarly promoted the MMP. However, HV-3 had no additional protection on the MMP in the presence of VCP siRNA (Fig. 4a), suggesting that VCP is required for HV-3 on improvement of mitochondrial function. Treatment with HV-3 also reduced the number of fragmented mitochondria (Fig. 4b) and increased mitochondrial length (Fig. 4c) in HdhQ111 cells. Using EM analysis, we further observed an increase in the number of mitophagosomes in HdhQ111 cells, whereas treatment with HV-3 reduced this accumulation (Fig. 4c). In HdhQ111 striatal cells subjected to 24 h of serum withdrawal, HV-3 treatment reduced the release of high mobility group box 1 (HMGB1) and lactate dehydrogenase (LDH), two indicators of cell death (Fig. 4d,e). We found that HV-3 treatment had no effect on the ER stress response (Supplementary Fig. 6a), excluding the possibility that the protection provided by HV-3 to mitochondria is a secondary consequence of the inhibition of ER stress.

Neurons derived from HD patient-induced pluripotent stem cells (HD-iPS cells) exhibited mitochondrial damage and increased cell death[5,41]. In neurons immunopositive for both anti-DARPP-32 (a marker of medium spiny neurons) and anti-Tubulin β-III (a marker of neurons), treatment with HV-3 reduced neurite shortening compared with patient neurons treated with control peptide TAT (Fig. 4f,g). The neuroprotective effects of HV-3 were consistently

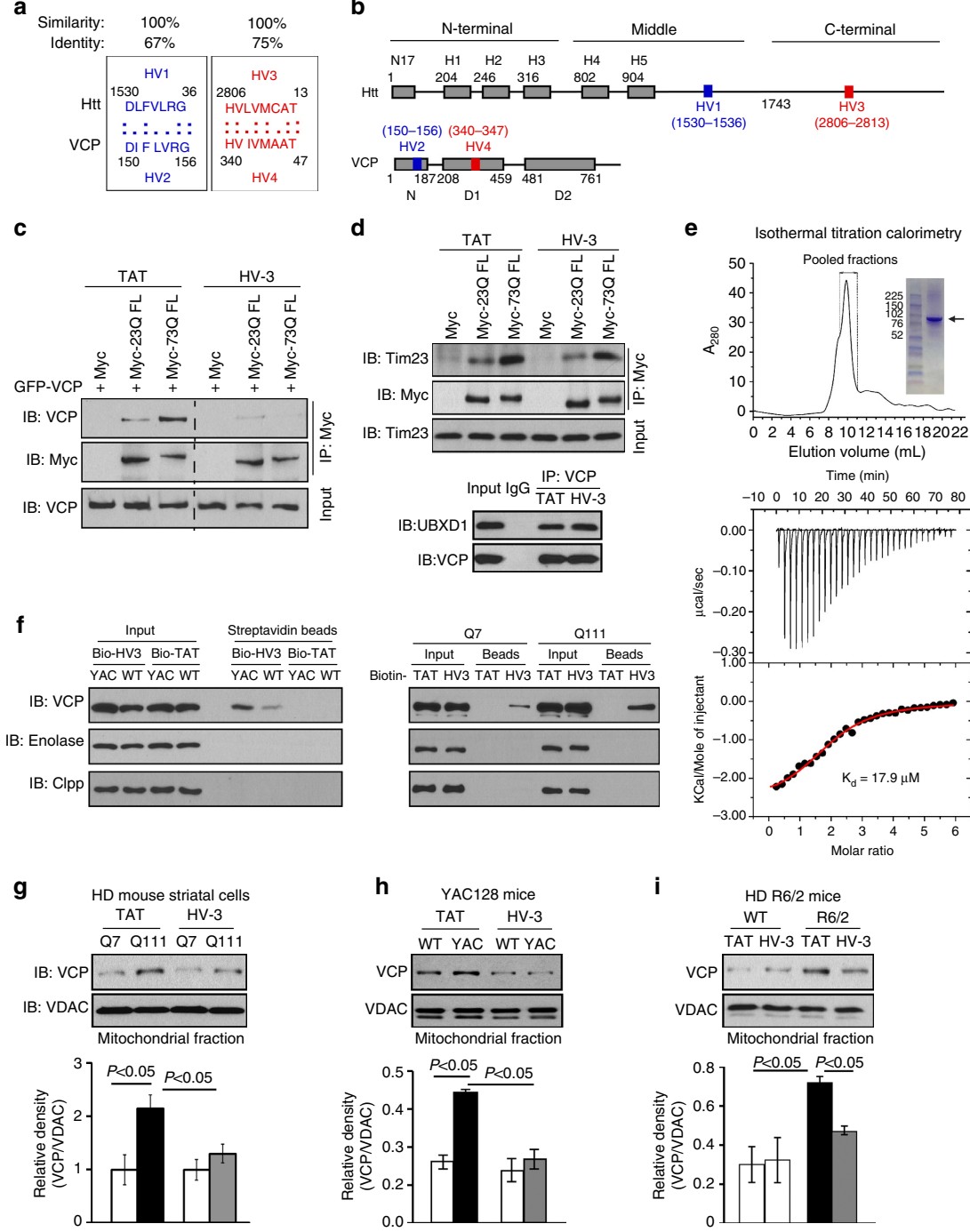

**Figure 3 | HV-3 peptide blocks Htt/VCP binding.** (**a**) Sequence of homology between VCP (human, AAI21795) and Htt (human, NP_002102). Amino acids are represented by the one-letter code; stars (*) indicate identical amino acids; Columns (:) indicate high similarity between amino acids. (**b**) Stick drawings of VCP and Htt main domains. Highlighted in the same colours are the two regions of homology between the two proteins, regions HV-1 and HV-3 in Htt and the corresponding regions HV-2 and HV-4 in VCP. (**c**) HEK293 cells were transfected with Myc-full-length Htt with 23 Q or 73Q (Myc-23Q FL or Myc-73Q FL) for 48 h following treatment with HV-3 or TAT (3 μM per day, each). The total lysates of cells were subjected to IP followed by WB with the indicated antibodies. (**d**) The total cell lysates were subjected to IP followed by WB in the indicated groups. (**e**) Gel filtration chromatogram and SDS-PAGE gel of recombinantly expressed and purified full-length mouse VCP/p97 (upper). Equilibrium binding isotherm for VCP titrated against HV-3 peptide at 15 °C (lower). Each downward spike is from a single injection of HV-3 into the sample cell. The heat exchanged during each injection is calculated from the area under the spike and fit to a binding isotherm. The $K_d$ and n for HV-3 binding are $17.9 \pm 7$ μM and $2.02 \pm 0.23$, respectively. The values of $\Delta H$ and $\Delta S$ are $-2.145 \pm 0.65$ Kcal mol$^{-1}$ and $13.97 \pm 3.4$ cal mol$^{-1}$ deg$^{-1}$, respectively. (**f**) Biotin-conjugated HV-3 or TAT (10 μM, each) was incubated with total lysates of HD cells or YAC128 mouse brains. Immunoprecipitates were analysed by WB with the indicated antibodies. All Blots shown above are representative of three independent experiments. (**g**) HD cells were treated with TAT or HV-3 (3 μM per day for 3 days), $n = 3$. (**h**) YAC128 or wild-type mice from 3–6 months of age and (**i**) R6/2 or wild-type mice from 5 to 9 weeks of age were received either TAT or HV-3 (3 mg kg$^{-1}$ per day), $n = 6$ mice/ group. VCP mitochondrial levels were determined by WB. Loading control: VDAC. Data are mean ± s.e.m. (**g–i**) ANOVA with Holm-Sidak *post hoc* test.

associated with improved MMP and mitochondrial length along neurites (Fig. 4h). Further, HV-3 treatment suppressed neuronal cell death in neurons subjected to growth factor withdrawal (Fig. 4i). Taken together, these results demonstrate that treatment with HV-3 protects against mitochondrial damage and cell death under HD-associated conditions.

We found that the peptide HV-3 had only minor effects on VCP mitochondrial levels, MMP, and mitochondrial morphology, as well as cell survival rate in wt counterparts of the above HD models (Figs 3 and 4). This is likely the result of less binding between VCP and wt Htt under basal conditions (Figs 2a and 3c). Normal and mutant polyglutamine proteins

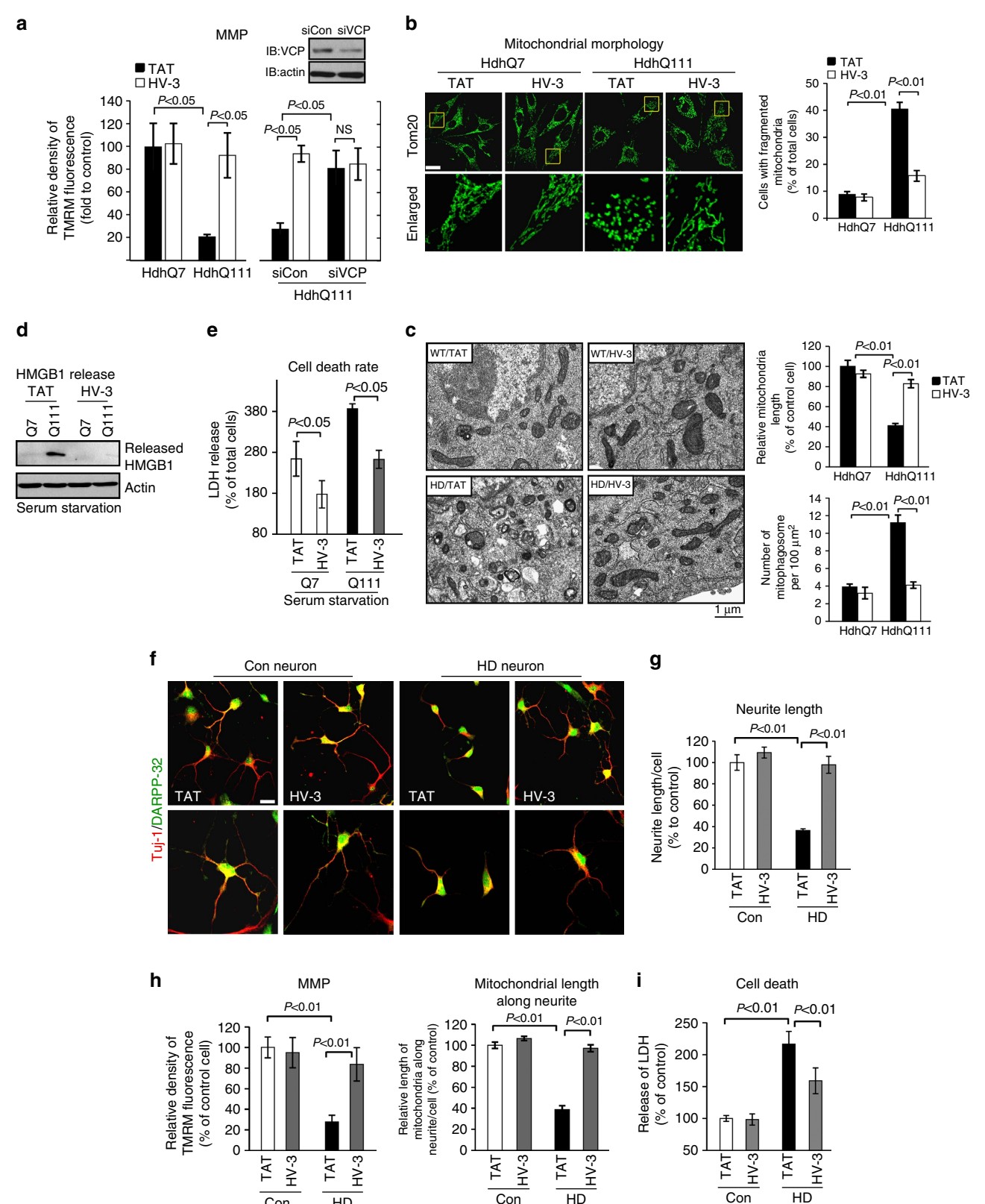

interact with VCP, but only mutant proteins specifically affect the activity of VCP and impair its function[26], thus it is also possible that disruption of wt Htt/VCP interaction by HV-3 results in only minor physiological impacts.

**VCP translocation to mitochondria impairs mitophagy in HD**. Apoptosis and autophagic cell death are manifested in HD neuropathology[42,43]. Blocking VCP recruitment to mitochondria by treatment with HV-3 did not affect apoptosis, as evaluated by the activity of caspase-3 (Supplementary Fig. 6b). In contrast, HV-3 treatment greatly reduced the accumulation of mitophagosomes in HdhQ111 cells (Fig. 4c). Down-regulation of VCP by VCP siRNA in HdhQ111 cells reduced the levels of mitochondria-associated LC3 II, which is a marker of mitophagy[44] (Fig. 5a). Expression of Flag-VCP in wt mouse striatal cells induced GFP-LC3B association with mitochondria, which could be inhibited by treatment with HV-3 (Fig. 5b). Note that HV-3 treatment had no effect on the total protein level of GFP-LC3B (Supplementary Fig. 6c). Thus, we speculated that mtHtt-induced VCP accumulation on mitochondria triggers mitochondria-associated autophagy.

In HdhQ111 cells, we observed an increased number of GFP-LC3B puncta, a specific marker for autophagosomes, and hyperactivity of lysosome enzyme Cathepsin B, both of which were reduced by treatment with HV-3 (Fig. 5c,d). Similarly, neurons derived from HD-iPS cells exhibited lower mitochondrial mass and lysosome hyperactivity, whereas treatment with HV-3 corrected these aberrant events (Fig. 5e,f). Further, we examined the ultrastructure of striatal mitochondria in YAC128 mice. Consistent with the findings in cells (Fig. 4c), we observed an increase in the number of mitophagosomes in 9-month-old YAC128 mice treated with the control peptide TAT, which was reduced by HV-3 treatment (Fig. 5g). These findings suggest that inhibition of VCP mitochondrial accumulation in HD by HV-3 suppress excessive mitophagy and improve mitochondrial quality.

LC3 in mammals or Atg8 in yeast plays a key role in both autophagosome membrane biogenesis and cargo recognition[45]. In yeast, Atg32 functions as a receptor on mitochondria to initiate mitophagy through interaction with Atg8 (refs 46,47). Similarly, mammalian mitophagic adaptors, such as FUNDC1 (ref. 48), p62 (ref. 49), BNIP3 (ref. 50) and AMBRA1 (ref. 51), all bind to LC3 via a typical linear motif with a core consensus sequence of W/Y/F xx L/I/V, also called LC3-interacting region (LIR)[52]. Given the above findings, we hypothesized that VCP might bind to LC3 on mitochondria to enhance mitophagosome production. Using an iLIR server[53], we found that VCP contains two segments of sequence (LEA**Y**RP**I**R and AVE**F**KV**V**E) located in the β stands of the N-terminal (Fig. 6a) that fulfil the characteristics of the LIR[52]. To determine whether VCP binds

to LC3 via putative LIR motifs, we generated two mutants (VCP-YI[AA] and VCP-FV[AA]) in which Y/I and F/V were all replaced by alanine, respectively. In HeLa cells co-expressing Myc-VCP and GFP-LC3B, we found that Myc-VCP was bound to GFP-LC3B in the mitochondrial fractions of cells (Fig. 6b). Expression of either VCP-YI[AA] or VCP-FV[AA] abolished the VCP/LC3 interaction (Fig. 6b). Moreover, expression of VCP-YI[AA] or VCP-FV[AA] reduced LC3 association with the mitochondria (Fig. 6c) and increased mitochondrial mass (Fig. 6d) compared to cells expressing VCP wt. These data demonstrate that mitochondria-accumulated VCP accelerates mitophagy by interacting with LC3 through the LIRs.

To determine direct consequences of VCP mitochondrial accumulation on mitophagy and cell survival, we generated a construct encoding VCP fused to a flag-vector containing a mitochondrial targeting sequence (MTS) (Flag-mtVCP). In HeLa cells expressing Flag-mtVCP, we confirmed the enrichment of Flag-mtVCP on mitochondria (Supplementary Fig. 7a). We further observed that expression of Flag-mtVCP induced a relocalization of the mitochondria network, forming mitochondrial aggregates around the perinuclear envelope (Supplementary Fig. 7a), which is an intermediate step of mitophagy[51,54]. The occurrence of mitochondrial aggregates in cells expressing Flag-mtVCP increased approximately sevenfold relative to cells not expressing Flag-mtVCP (Supplementary Fig. 7b). Moreover, the presence of Flag-mtVCP in cells decreased MMP and mitochondrial mass (Supplementary Fig. 7c,d), but induced an increase in the percentage of GFP-LC3B colocalizing with Tom20-labelled mitochondria (Supplementary Fig. 7e). On treatment with bafilomycin A to prevent autophagosome-lysosome fusion, Flag-mtVCP expression elevated the autophagic flux of the mitochondria (Supplementary Fig. 7f), indicating an increased rate of mitochondrial degradation.

In rat primary striatal neurons, expression of Flag-mtVCP-WT elicited mitochondrial aggregates and caused neurite shortening in medium spiny neurons that were labelled by anti-DARPP-32 antibody (Fig. 6e,f-top panel, g,h). In contrast, neurons expressing either Flag-mtVCP-FV[AA] or Flag-mtVCP-YI[AA] exhibited fewer mitochondrial aggregates and longer neurites of medium spiny neurons (Fig. 6e,f-middle and bottom panels, g,h). Thus, mitochondria-accumulated VCP contributed to mitochondrial and neuronal damage in primary striatal neurons via impairment of the mitophagic process.

**HV-3 treatment reduces behavioural phenotypes of HD mice**. We next examined whether blocking VCP accumulation on the mitochondria provides neuroprotection in *in vivo* animal models of HD.

---

**Figure 4 | HV-3 treatment reduces mitochondrial damage and cell death in HD cell cultures.** Mouse HdhQ7 and HdhQ111 striatal cells were treated with control peptide TAT or peptide HV-3 (3 uM/day for 3 days). (**a**) Left panel: Mitochondrial membrane potential (MMP) was determined by TMRM fluorescent dye. Right panel: HdhQ111 cells were transfected with control siRNA (siCon) and VCP siRNA (siVCP) for three days. The MMP was determined by TMRM in HdhQ111 cells treated with TAT or HV-3. (**b**) Mitochondrial morphology was determined by staining cells with anti-Tom20 antibody. Scale bar: 10 μm. The percentage of cells with fragmented mitochondria relative to the total number of cells was quantitated. At least 100 cells per group were counted. (**c**) Mitochondrial morphology was determined by EM. The length of mitochondria and the number of mitophagosomes were quantitated. At least 90 mitochondria per group were counted. (**d**) HD striatal cells were subjected to serum starvation for 24 h. HMGB1 release into culture medium was determined by IB analysis with anti-HMGB1 antibody. (**e**) HD striatal cells were subjected to serum starvation for 24 h. Cell death was determined by the release of LDH. Control and HD patient-iPS cell derived neurons were treated with HV-3 or TAT at 1 μM per day for 5 days starting 30 days after initiation of neuronal differentiation. (**f**) Left: Neurons were stained with anti-DARPP-32 and anti-Tuj-1 antibodies to indicate medium spiny neurons. Upper: a cluster of neurons; lower: individual neurons. Scale bar: 10 μm. (**g**) Quantitation of neurite length of medium spiny neurons. At least 50 neurons per group were counted by an observer blind to experimental conditions. (**h**) Left: the MMP was determined by TMRM fluorescent dye. Right: Mitochondria were stained by anti-Tom20 antibody. Mitochondrial length along neurites of DARPP32/Tuj1-positive neurons was quantitated. (**i**) Neuronal cell death induced by the withdrawal of the growth factor BDNF for 24 h was determined by the release of LDH. All data are mean ± s.e.m. from at least three independent studies. ANOVA with Holm-Sidak *post hoc* test.

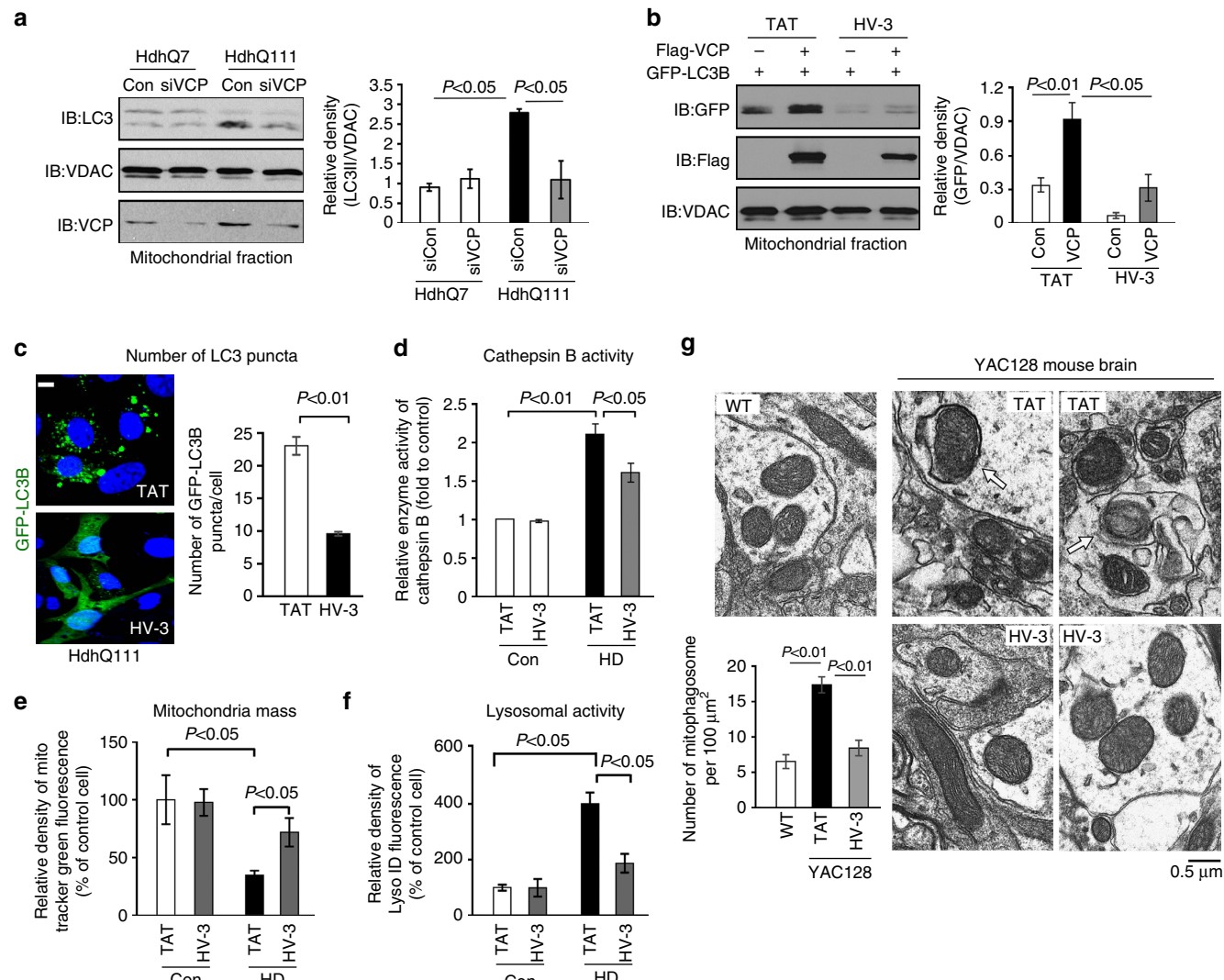

**Figure 5 | Treatment of HV-3 reduces excessive mitophagy in HD cell cultures and HD mouse brains.** (**a**) HdhQ7 and HdhQ111 cells were treated with control siRNA (con) or VCP siRNA (siVCP) for 48 h. Mitochondria were isolated and LC3 mitochondrial levels were determined by WB. Representative blots are from three independent experiments. The quantitation of LC3 II levels on mitochondria is provided on the right. VADC was used as a loading control. (**b**) Flag-VCP and GFP-LC3B were co-transfected into wild-type striatal cells. Mitochondria were isolated after 36 h of transfection. The GFP-LC3B levels on mitochondria were examined by WB. VDAC was used as a loading control. Representative blots are from three independent experiments. Histogram: quantitation of GFP-LC3B mitochondrial protein level. HdhQ7 and HdhQ111 cells were treated with control peptide TAT or peptide HV-3 (3 uM/day for 3 days). (**c**) HdhQ111 cells were transfected with GFP-LC3B for 24 h. The number of GFP-LC3B puncta was quantitated and shown in the histogram. Scale bars: 10 μm. (**d**) Enzyme activity of lysosomal Cathepsin B was measured using a Cathepsin B assay kit. Control and HD patient-iPS cell derived-neurons were treated with HV-3 or TAT at 1 μM per day for 5 days starting 30 days after neuronal differentiation. (**e**) Mitochondrial mass was measured by the fluorescent density of Mitotracker green. (**f**) Lysosomal activity was examined by staining neurons with Lyso-ID Red dye. (**g**) YAC128 mice and wild-type mice were treated with TAT or HV-3 (3 mg kg$^{-1}$ per day) from the age of 3–9 months. EM analysis of striata from 9-month-old wild-type and YAC128 mice was performed. Arrows indicate mitophagosomes. Histogram: the number of mitophagosomes per 100 μm$^2$ was counted and quantitated. Fifteen random areas in the striatum of each animal were analyzed. All the data are mean ± s.e.m. of three independent experiments. ANOVA with Holm-Sidak *post hoc* test.

HD R6/2 mice treated with the control peptide TAT exhibited decreased horizontal and vertical activities as well as less total distance travelled in the test of spontaneous locomotion when evaluated at the age of 13 weeks. Treatment with HV-3 markedly corrected these motor deficits (Fig. 7a). The severity of clasping behaviour in R6/2 mice treated with HV-3 was significantly lower over the 4-week observation period than it was in those treated with the control peptide TAT (Fig. 7a). HV-3 treatment also resulted in increased body weight and survival rate of R6/2 mice (Fig. 7b,c). The treatment had no effects on motor ability, body weight, or life span in wt mice (Fig. 7a–c), suggesting a lack of toxicity of HV-3 treatment.

YAC128 mice exhibited progressively deficits in motor activities; they showed gradually decreasing motor coordination activity on the rotarod and defects in general motility measured by locomotor activity chambers. Sustained treatment with HV-3 improved general movement activity and rotarod performance of YAC128 mice starting at the age of 6 months, and the protection lasted until the age of 12 months (Fig. 7d,e). Again, the treatment did not affect motor activity in wt mice from 3 to 12 months of age.

We found that HV-3 at 3 mg kg$^{-1}$ per day was not toxic to naive mice (Supplementary Fig. 8). Treatment with HV-3 had no significant effects on the immunodensity of CD3, a marker of T-cell for adaptive immune response, in brain and spleen samples

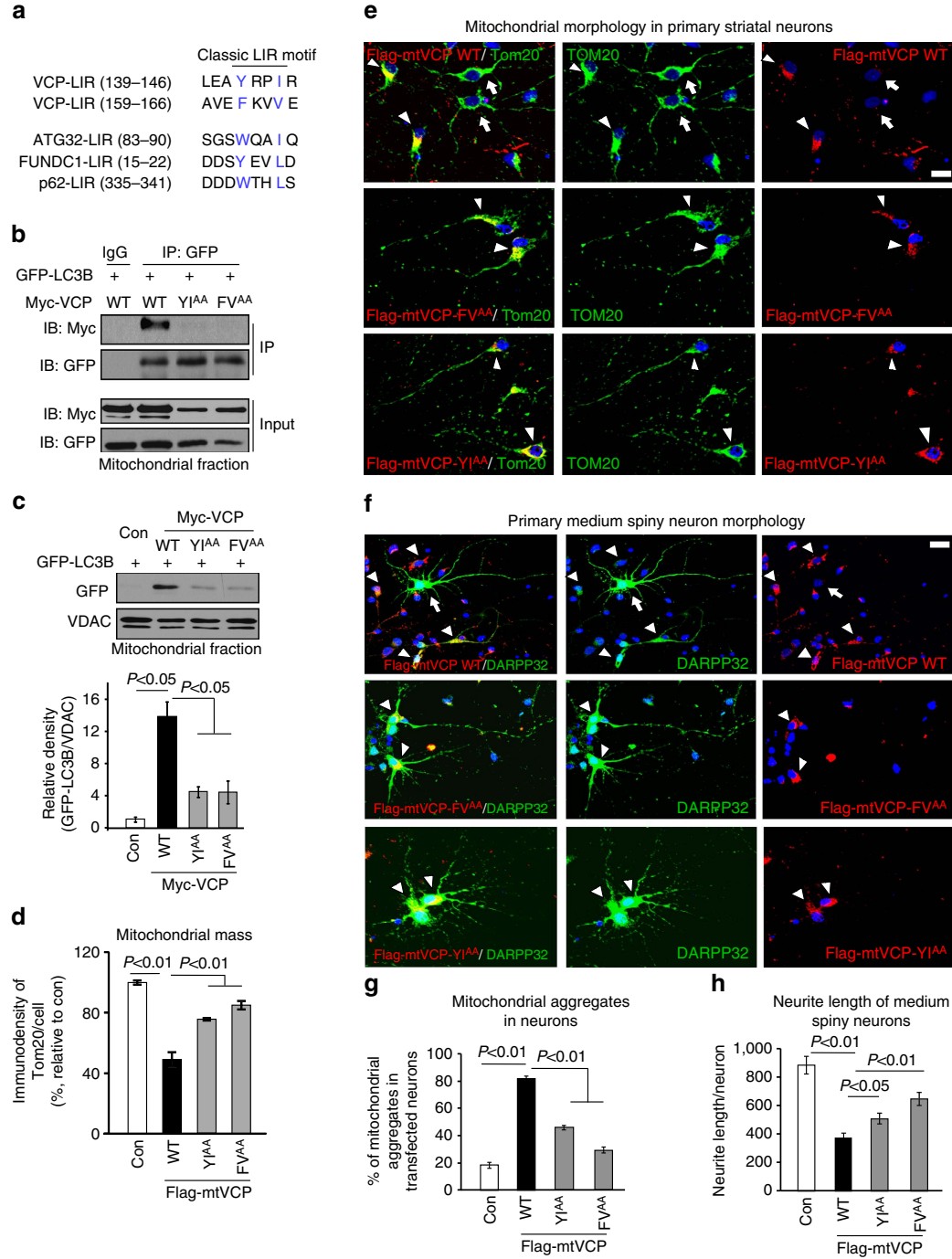

**Figure 6 | VCP causes excessive mitophagy by binding to LC3 via LIRs.** (**a**) Putative LIR sequences in VCP were aligned manually for comparison with the classical LIR motifs of ATG32, FUNDC1 and p62. The amino acids in blue indicate the conserved core residues of LIR. (**b**) GFP-LC3B was co-expressed with the indicated plasmids in HeLa cells. Mitochondrial lysates were subjected to IP with anti-GFP antibody, and immunoprecipitates were analysed by WB with anti-Myc and anti-GFP antibodies. Representative blots are from three independent experiments. (**c**) GFP-LC3B was co-transfected with the indicated plasmids in HeLa cells. Mitochondria were isolated and GFP-LC3B mitochondrial protein levels were determined by WB. Data are mean ± s.e.m. from four independent studies. (**d**) HeLa cells were transfected with the indicated plasmids. Mitochondria were stained with anti-Tom20 antibody. Mitochondrial mass was determined by quantitating fluorescent density of Tom20 immunostaining. At least 100 cells per group were counted. Data are mean ± s.e.m. from three independent studies. Primary rat striatal neurons (DIV 7) were transfected with either Flag-mtVCP-WT, or Flag-mtVCP-FV[AA] or Flag-mtVCP-YI[AA] plasmids for 3 days. (**e**) Neurons were stained with anti-Tom20 (green) and anti-Flag (red) antibodies. Mitochondrial morphology was examined by microscopy. (**f**) Medium spiny neurons were labelled with anti-DARPP-32 (green). Arrows indicate the cells that were not transfected with Flag-VCP. Arrowheads show the cells with transfected Flag-VCP. (**g**) Mitochondrial aggregates in neurons expressing Flag-mtVCP or Flag-mtVCP-FV[AA] or Flag-mtVCP-YI[AA] were quantitated. (**h**) Neuronal morphology was imaged and the neurite length of medium spiny neurons expressing Flag-mtVCP or Flag-mtVCP-FV[AA] or Flag-mtVCP-YI[AA] was quantitated. At least 50 neurons per group were counted by an observer blind to experimental conditions. Scale bars: 10 µm. All the data are mean ± s.e.m. from three independent experiments. ANOVA with Holm-Sidak *post hoc* test.

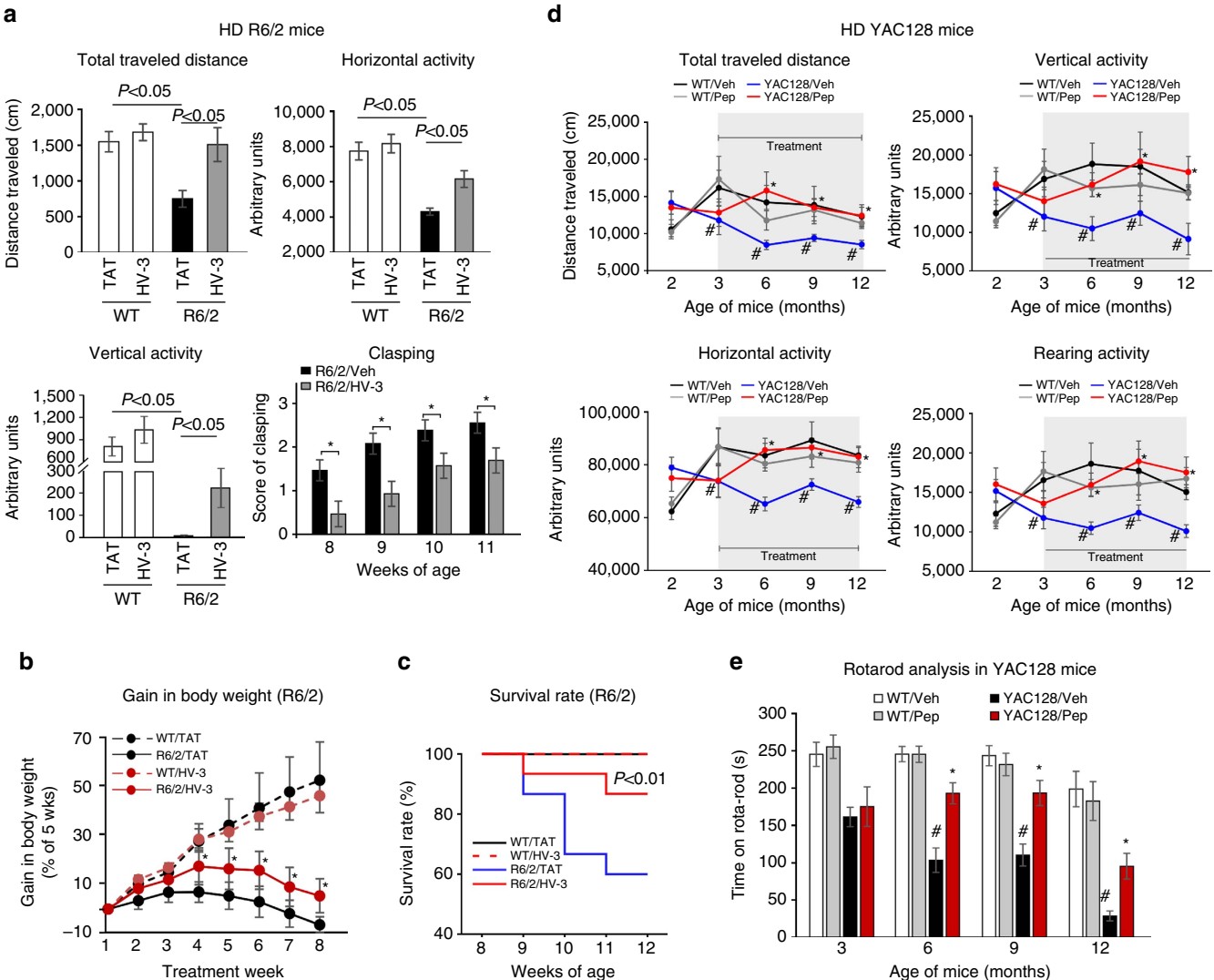

**Figure 7 | HV-3 treatment reduces motor deficits in both R6/2 and YAC128 HD mice.** HD R6/2 mice and wild-type littermates were treated with either the control peptide or peptide HV-3 (at 3 mg kg[−1] per day, subcutaneous administration with an Alzet osmotic pump) from 5 to 13 weeks of age (see treatment timeline in Supplementary Fig. 5a). (**a**) One hour of overall movement activity in R6/2 mice and wild-type littermates (total travelled distance, horizontal and vertical activities) was determined by locomotion activity chamber at the age of 13 weeks ($n = 15$ mice per group). ANOVA with Holm-Sidak *post hoc* test. Hindlimb clasping was assessed with the tail suspension test once a week from the ages of 8 to 11 weeks ($n = 15$ mice per group). *$P < 0.05$ (Paired Student's *t*-test). Body weight (**b**) and survival (**c**) were recorded from the age of 5–13 weeks ($n = 15$ mice per group). *$P < 0.05$ versus HD mice treated with control peptide TAT. YAC128 mice and wild-type littermates were treated with the TAT or HV-3 peptides from the age of 3 to 12 months. Mouse behavioural and HD-associated pathology were determined every three months after beginning treatment (See treatment timeline in Supplementary Fig. 5a) (**d**) 24 h of general motility of YAC128 mice and wild-type littermates was monitored by a locomotion activity chamber at the indicated age ($n = 15$–20 mice per group). #$P < 0.05$ versus wild-type mice treated with TAT; *$P < 0.05$ versus HD mice treated with TAT. (**e**) Rotarod performance of YAC128 and wild-type mice was evaluated at the indicated age ($n = 15$–20 mice per group). #$P < 0.05$ versus wild-type mice treated with TAT; *$P < 0.05$ versus HD mice treated with TAT. (**b**–**e**) Repeated-measures two-way ANOVA with Bonferroni's *post-hoc* test. All data are expressed as mean ± s.e.m.

from naive mice, and had no obvious effects on the size and weight of the spleens in naive mice (Supplementary Fig. 8). These results suggest that HV-3 might be safe for use in animals.

**HV-3 treatment reduces neuropathology of HD mice.** The levels of dopamine signalling protein, DARPP-32, enriched in medium spiny neurons are decreased in the striatum of HD patients and mouse models[55]. Thus, DARPP-32 has been used as a marker to assess neuronal degeneration in HD mouse models. Western blot analysis of striatal extracts revealed a significant reduction of DARPP-32 protein levels in both R6/2 and YAC128 mice. HV-3 treatment significantly increased DARPP-32 levels in the two mouse models (Fig. 8a). In HD R6/2 mice, we consistently

observed a decrease in the area occupied by DARPP-32-immunostained cells in the striatum, which was increased by HV-3 treatment (Fig. 8b,c). To further assess whether HV-3 treatment can suppress neurodegenerative pathology in HD, we conducted unbiased stereology analyses to measure the number of striatal neurons in YAC128 mice at the age of 12 months. We found that treatment with HV-3 significantly increased the number of neurons positive for anti-NeuN immunostaining in the dorsolateral striatum (Fig. 8d).

**Discussion**
In this study, we reported that mtHtt-induced recruitment of VCP to mitochondria caused HD-associated neurodegeneration,

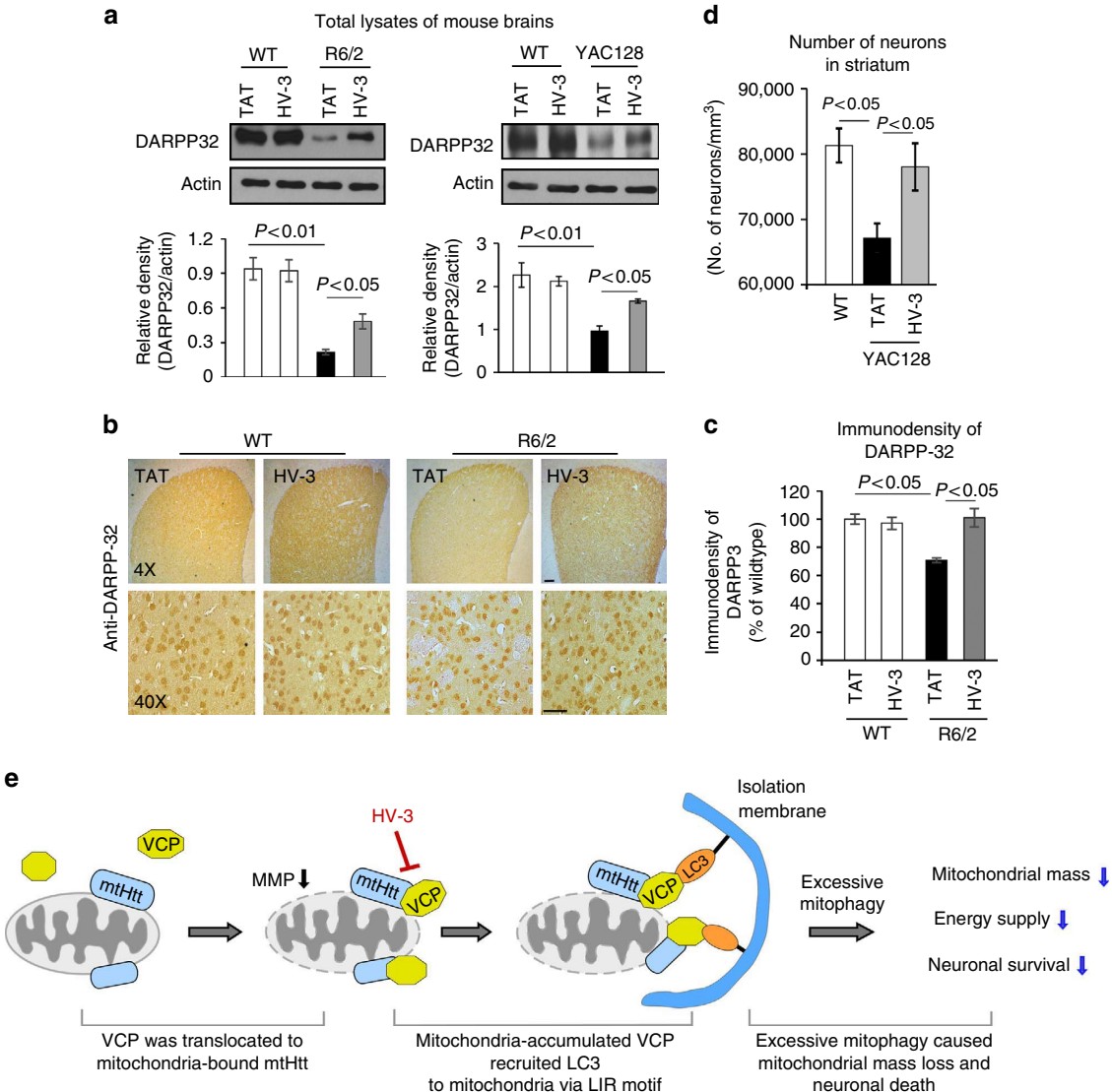

**Figure 8 | HV-3 treatment reduces mitochondrial defects and neuropathology in HD mice.** (**a**) DARPP-32 protein levels were determined by WB of R6/2 (left) and YAC128 (right) mouse striatal extracts. Upper: representative IB; Lower: histogram of quantification of DARPP-32 levels. Actin was used as a loading control. Data are mean ± s.e.m. $n = 6$ mice/group. (**b**) Photomicrographs of DARPP-32 immunostaining were obtained from the dorsolateral striatum of TAT- or HV-3-treated R6/2 mice. Scale bar: 100 μm. (**c**) Quantitation of DARPP-32 immunodensity. Data are mean ± s.e.m. $n = 6$ mice/group. (**d**) Quantitation of NeuN-immunopositive cells in the dorsolateral striatum. Data are mean ± s.e.m. $n = 6$ mice/group. (**a,c,d**) ANOVA with Holm-Sidak *post hoc* test. (**e**) A summary scheme. VCP is selectively recruited to the mitochondria by interacting with mitochondria-bound mtHtt. Mitochondria-accumulated VCP acts as a mitophagic adaptor to bind to the autophagosome component LC3 via an LC3-interacting region (LIR motif). As a result, mtHtt-induced VCP association with mitochondria causes excessive mitophagy which results in mitochondrial mass loss, mitochondrial dysfunction and neuronal cell death. Blocking mtHtt to VCP binding on mitochondria by a selective peptide HV-3 inhibits VCP mitochondrial accumulation, which reduces excessive mitophagy and subsequent neuronal degeneration. Consequently, treatment with HV-3 both in HD cultures and in HD animals reduces HD-associated neuropathology.

as evidenced by the fact that blocking VCP mitochondrial accumulation with the peptide HV-3 corrected excessive mitophagy and mitochondrial dysfunction, and reduced HD neuropathology *in vitro* and *in vivo* (Fig. 8e). Thus, the mitochondria-accumulated VCP might represent a new therapeutic target for combatting neurodegeneration in HD.

VCP has been shown to bind to Htt in HD mouse brains and in postmortem patient brains[26,30]. Here, we further showed that the binding of VCP and mtHtt mainly occurred on the mitochondria of HD cell cultures and animal brains. It is possible that VCP was aberrantly recruited to mitochondria via mitochondria-bound mtHtt through protein–protein interactions. Indeed, our *in vitro* and *in vivo* data showed that blocking VCP/mtHtt binding with HV-3 abolished VCP translocation to mitochondria and reduced

mitochondrial damage, further emphasizing that the binding of VCP/mtHtt is required for VCP relocation to the mitochondria. Significantly, inhibition of VCP/mtHtt binding reduced HD-related behavioural and pathological phenotypes in two HD transgenic mouse models. Thus, the formation of the aberrant VCP/mtHtt complex on the mitochondria may be a key step in initiating mitochondrial injury, which in turn results in the neuronal pathology of HD.

The peptide HV-3 is derived from Htt and represents a sequence homologous to VCP. We further showed that HV-3 binds to VCP; HV-3 has a relatively high affinity for VCP assessed by the ITC and biotin-HV-3 specifically pulled down VCP (Fig. 3). These findings suggest that HV-3 might compete with Htt binding to VCP or that it targets VCP and prevents the

exposure of the VCP-binding site on Htt. Fujita et al showed that mtHtt weakly affects VCP enzyme activity[26], suggesting that the enzyme activity of VCP might not be a key in mediating the binding of VCP to mtHtt. Currently we are determining whether HV-3 affects VCP conformational changes using VCP crystal structures, which may alter VCP activity.

We found that HV-3 can block VCP accumulation on mitochondria and provide neuroprotection in HD R6/2 mice in which an N-terminal mtHtt fragment is expressed. VCP has been shown to bind to Htt exon 1 fragment with expanded polyQ via the polyQ tract sequence and co-localize with mtHtt in perinuclear cytoplasmic region of neurons in R6/2 mice[26]. Because N-terminal mtHtt fragments can co-localize and co-aggregate with normal Htt fragments[56], it is possible that VCP, mtHtt fragment and endogenous Htt form a protein complex in HD R6/2 mice. HV-3 may block VCP accumulation on the mitochondria in R6/2 mice by disrupting the interaction of the complex, thus reducing subsequent mitochondrial damage. However, other mechanisms may exist, which remain to be further investigated.

VCP translocation to mitochondria mediates turnover of the mitochondrial fusion protein Mitofusin 1 and subsequent Parkin-related mitophagy in HeLa cells[15]. In our study, we found that targeting VCP to mitochondria is sufficient to induce massive mitochondrial clearance in HeLa cells and HD striatal cells when Parkin is not present (Supplementary Fig. 7g). Thus, VCP-mediated excessive mitophagy in HD may be independent of the Parkin-related pathway, although the detailed mechanism requires further investigation. While a basal level of mitophagy is essential for neuronal health, excessive mitophagy may cause harm by removing too many mitochondria, which leads to rapid neuronal death[44,57] and is centrally implicated in the pathogenesis of neurodegenerative diseases[58]. Because mtHtt is ubiquitously expressed and is required for VCP translocation to mitochondria (Fig. 1c), it is conceivable that mtHtt causes extensive accumulation of VCP on the mitochondria, which disrupts the balance of mitophagy, leading to excessive mitochondrial degradation and subsequent neuronal death.

Htt can function as a scaffold protein for autophagy, and C-terminal Htt has been suggested to be essential for mitophagy induction under physiological conditions[59,60]. Expression of the C-terminal Htt fragment in rat primary cortical neurons or striatal cells caused cell death[59,61]. Similarly, we found that targeting VCP to mitochondria is required for mitophagy and leads to death of medium spiny neurons, again supporting the idea that well-balanced mitophagy is critical for maintaining neuronal survival. Although there are recent findings that mtHtt impairs macroautophagy by negatively regulation of ULK-mTOR pathway[60] and GAPDH-mediated mitophagy[62], we found here that VCP uses LIR motifs to interact with LC3 following mitophagy induction, the process of which was accelerated by the presence of mtHtt. The Htt protein also carries a number of LIR motifs in the c-terminal of the protein[59]. It remains to be determined if mtHtt and VCP dependently or independently transduce mitophagic signalling in HD. In addition, PGC1α, a key regulator of mitochondrial biogenesis implicated in HD pathogenesis, has recently been reported to regulate mitophagy and autophagy through TFEB signalling[63,64]. It would be interesting to determine whether there is cross talk between PGC1α and VCP-mediated mitophagy in HD.

In this study, we found that subcutaneous treatment of HV-3, for 8 weeks in HD R6/2 mice and for 9 months in YAC128 mice, reduced behavioural abnormalities and increased neuronal survival, further supporting our hypothesis that VCP recruitment to mitochondria by mtHtt is an initial step for the induction of neuronal degeneration in HD. Moreover, we found that

the peptide HV-3 is well tolerated in normal mice. Thus, development of inhibitors, such as HV-3-like reagents, may have the potential to open up a new therapeutic route for HD and multiple polyglutamine diseases in which VCP translocation to mitochondria is characterized.

## Methods

**Antibodies and reagents.** Protein phosphatase inhibitor and protease inhibitor cocktails were purchased from Sigma-Aldrich. VCP inhibitor Eer I and proteasome inhibitor MG132 were from Tocris Bioscience. Antibodies for Tom20 (sc-11415, 1:1,000), c-Myc (sc-40, 1:1,000), GFP (sc-9996, 1:1,000), GST (sc-138, 1:500), CD3 (sc-20047, 1:500), Enolase (sc-15343, 1:1,000), Tim23 (sc-514463, 1:500) and Parkin (sc-32282, 1:1,000) were from Santa Cruz Biotechnology. Full-length Htt (MAB2166, 1:1,000), polyQ (MAB1574, 1:1,000), EM48 (MAB5374, 1:1,000) and NeuN (MAB377, 1:500) antibodies were from Millipore. Pan-actin (A1978, 1:10,000) and Flag (F3165, 1:5,000) antibodies were from Sigma-Aldrich. Antibodies for VDAC (ab14734, 1:2,000), Clpp (ab124822, 1:1,000), UBXD1 (ab103651, 1:500) and VCP (ab109240, 1:10,000) were from Abcam. EEA1 (3288, 1:500) and LC3 (2775, 1:1,000) antibodies were from Cell Signalling, WFS1 (NB100-1918, 1:1,000) antibody was from Novus, HMGB1 (10829-1-AP, 1:1,000) antibody was from Proteintech, and GRP78 (ADI-SPA-826, 1:1,000) and Calnexin (ADI-SPA-860, 1:1,000) antibody was from Enzo Life Sciences. Anti-mouse IgG and anti-rabbit IgG, peroxidase-linked, species-specific antibodies were from Thermo Scientific.

**Constructs and transfection.** Myc-tagged full-length Htt with 23Q or 73Q plasmid was obtained from the CHDI foundation. The full-length VCP wt and GFP-LC3B plasmids were obtained from Addgene. To construct the mitochondria-targeting VCP plasmid, CMV-mito-GEM-GECO1 was digested with BamHI and HindIII, and VCP was PCR-amplified and inserted into the plasmid backbone. Site mutation of the VCP plasmid was performed using a site-mutagenesis kit (Agilent Technologies, Inc.). Cells were transfected with TransIT-2020 (Mirus Bio, LLC) following the manufacturer's protocol.

**Cell culture.** Immortalized striatal cell lines HdhQ111 mutant and HdhQ7 wt cells were obtained from the CHDI Foundation. Cells were cultured in DMEM supplemented with 10% FBS, 100 μg/ml penicillin, 100 μg ml$^{-1}$ streptomycin, and 400 μg ml$^{-1}$ G418. Cells were grown at 33 °C in a 5% $CO_2$ incubator. Cells within 14 passages were used in all experiments.

Human cervix carcinoma cells (HeLa cells) and HEK293 cells were maintained in DMEM supplemented with 10% FBS and 1% (v/v) penicillin/streptomycin.

HD patient fibroblasts (HD1: GM21756; HD2: GM03621; purchased from Coriell Institute, USA) and normal fibroblasts (Con 1, fibroblasts from adult, HDFa; Con 2, fibroblasts from juvenile, HDFn; purchased from Invitrogen) were maintained in MEM supplemented with 15% (vol/vol) FBS and 1% (vol/vol) penicillin/streptomycin.

Primary striatal neurons from E18 rat midbrain tissue (BrainBits, Springfield, IL, USA) were seeded on cover slides that were coated with poly-D-lysine/laminine and grown in neurobasal medium supplemented with 2% B27 and 0.5 mM glutamate. At 7 DIV, cells were transfected with the control vector or flag-VCP$^{mt}$ using TransIT-2020 Transfection Reagent combined with formulated BrainBits transfection media for primary neurons (BrainBits, USA).

iPS cells from a normal subject and a HD patient (carrying 41 CAG repeats) were differentiated into neurons using the protocol from our previous studies[5]. Briefly, iPS cells were plated onto 6-well plates precoated with 2.5% Matrigel and allowed to reach 90% confluence in feeder-free medium. For the first 10 days, cells were treated with SB431542 (10 μM; Tocris Bioscience) and Noggin (100 ng/ml) in Neural Media (NM) containing Neurobasal and DMEM (1:1), B27 supplement minus vitamin A (50×, Invitrogen), N2 supplement (100×, Invitrogen), GlutaMax (Invitrogen, 100×), FGF2 (20 ng μl$^{-1}$) and EGF (20 ng μl$^{-1}$), 100 units per ml penicillin and 100 μg ml$^{-1}$ streptomycin). For the next 10 days, cells were treated with human recombinant Sonic hedgehog (SHH, 200 ng/ml), DKK1 (100 ng/ml) and BDNF (20 ng/ml) and 10 μM Y27632 (Sigma) in neuronal differentiation medium containing Neurobasal and DMEM (1:3), B27, N2, GlutaMax and PS. Cells were then switched to treatment with BDNF (20 ng ml$^{-1}$), ascorbic acid (200 μM, Sigma-Aldrich), cAMP (0.5 mM, Sigma-Aldrich) and Y27632 (10 μM) in neuronal differentiation medium. All growth factors were purchased from PeproTech (Rocky Hill, NJ, USA). Twenty days after the initiation of differentiation, neurons (about 5,000 cells) were plated onto 12-mm poly-D-lysine/laminine-coated coverslips and grown in 24-well plates in neuronal differentiation medium.

All of the above cells were maintained at 37 °C in 5% $CO_2$–95% air.

**RNA interference.** For silencing Htt and VCP in HD striatal cells, control siRNA, mouse Htt and mouse VCP siRNA were purchased from Thermo Fisher Scientific. HdhQ7 and HdhQ111 cells were transfected either with control siRNA, or Htt or VCP siRNA using TransIT-TKO Transfection Reagent (Mirus Bio, LLC), according to the manufacturer's instructions. The sequences for the siRNAs used in

this study are as follows: mouse Htt, 5′-GGUUUAUGAACUGACUUUGTT-3′; Mouse VCP, 5′-GAGAGCAACCUUCGUAAG-3′; control non-targeting siRNA, 5′-TTCTCCGAACGTGTCACGT-3′.

**Isolation of subcellular fractions.** Cells were washed with cold PBS and incubated on ice for 30 min in a lysis buffer (250 mM sucrose, 20 mM HEPES-NaOH, pH 7.5, 10 mM KCl, 1.5 mM MgCl$_2$, 1 mM EDTA, protease inhibitor cocktail and phosphatase inhibitor cocktail). Mouse brains were minced and homogenized in the lysis buffer and then placed on ice for 30 min. Collected cells or tissue were disrupted 20 times by repeated aspiration through a 25-gauge needle, followed by a 30-gauge needle. The homogenates were spun at 800 g for 10 min at 4 °C, and the resulting supernatants were spun at 10,000 g for 20 min at 4 °C. The pellets were washed with lysis buffer and spun at 10,000 g again for 20 min at 4 °C. The final pellets were suspended in lysis buffer containing 1% Triton X-100 and were mitochondrial-rich lysate fractions. The supernatant was centrifuged at 100,000 g, 4 °C, for 1 h, the pellets were suspended in lysis buffer containing 1% Triton X-100 as ER fractions. The final supernatant was cytosolic fractions. The mitochondrial proteins VDAC, the ER protein WFS1 and the cytosolic protein Enolase were used as loading controls for mitochondrial, ER and cytosolic fractions, respectively.

**Immunoprecipitation.** Cells were lysed in a total cell lysate buffer (50 mM Tris-HCl, pH 7.5, 150 mM NaCl, 1% Triton X-100, and protease inhibitor) or in a mitochondrial isolation buffer above. Total or mitochondrial lysates or the mixture of ER and cytosolic fractions were incubated with the indicated antibodies overnight at 4 °C followed by the addition of protein A/G beads for 1 h. Biotin-HV-3 and Biotin-TAT (10 µM, each) were incubated with total lysates of cell cultures or mouse brains for overnight at 4 °C followed by the incubation with streptavidin beads for 1 h. Immunoprecipitates were washed four times with cell lysate buffer and were analysed by SDS–PAGE and IB.

**Rational design of peptide inhibitor.** Two nonrelated proteins that interact in an inducible manner have often shared short sequences of homology that represent sites of both inter- and intra-molecular interactions[36,65]. Similar to the peptide design for PKC peptide deltaV1-1 (ref. 37) and Drp1 peptide P110 (ref. 36), we used L-ALIGN sequence alignment software and identified two different regions of homology between VCP (VCP, Human, AAI21795) and Htt (Htt, human, NP_002102). These regions are marked as regions HV from 1 to 4. We found that all the homologous sequences are conserved in a variety of species including human, mouse, rat and fish. We synthesized the four peptides at American Peptide Company (Sunnyvale, CA) corresponding to regions HV-1-4 and conjugated them to the cell permeating TAT protein-derived peptide, TAT$_{47-57}$. Note that TAT$_{47-57}$-based delivery was used in culture and in vivo and was found to be safe and efficacious for delivery of peptide cargoes to cells and also to cross the blood-brain-barrier[5,36]. These peptides are referred to as HV-1, HV-2, HV-3 and HV-4. The purity was assessed as >90% by mass spectrometry. Lyophilized peptides were dissolved in sterile water and stored at −80 °C until use.

**VCP expression and purification.** The full-length mouse VCP/p97 (residues 2–806) was expressed with an N-terminal 6xHis-tag (Addgene: plasmid # 12373) in the Rosetta (DE3) strain of *Escherichia coli* (Novagen) as previously described[66]. Briefly, the cells were grown in LB media supplemented with Kanamycin and induced at OD∼1.0 for 16 h at 18 °C. Cells were homogenized in Buffer A (500 mM KCl, 25 mM Hepes pH 8.0, and 2 mM β-mercaptoethanol) in the presence of EDTA-free Complete Protease Inhibitor (Roche) and 2 mM PMSF. The lysate was cleared by centrifugation at 45,000 g for 1 h at 4 °C. Supernatant was incubated with 2 ml of Ni-NTA agarose resin (Qiagen) for 2 h at 4 °C. The resin was washed with Buffer A containing 50 mM imidazole and eluted with Buffer A containing 400 mM imidazole. VCP protein was further purified by passing through Superdex 200 increase 10/300 GL (GE Healthcare) size exclusion column equilibrated with Buffer B (250 mM KCL, 25 mM Hepes pH 8.0, 2 mM β-mercaptoethanol and 1 mM MgCl$_2$). The fractions containing the pure sample, as determined by SDS–PAGE analysis, were pooled and concentrated to a final concentration of ∼3–5 mg ml$^{-1}$.

**Isothermal titration calorimetry.** Binding of HV-3 peptide to VCP was measured by isothermal titration calorimetry on a MicroCal ITC200 (GE Healthcare). Before the calorimetric titrations, the protein was exchanged into Buffer C (100 mM NaCl, 10 mM Hepes, pH 8.0) on a PD-10 column (Amersham Biosciences) and concentrated to ∼5–6 mg ml$^{-1}$. The final concentration of protein in the cell (∼60–70 µM) was determined on a NanoDrop Spectrophotometer (Thermo Scientific NanoDrop 2000) using a molecular weight of 89.32 KDa and a molar extinction coefficient of 36.62 M$^{-1}$ cm$^{-1}$. The HV-3 peptide was dissolved in Buffer C to a final concentration of 2 mM and loaded into the syringe. The measurements were made at a constant cell temperature of 15 °C and repeated at least three times. Thirty successive injections of 1.2 µl each were titrated into the cell with constant stirring at 1,000 r.p.m. An equilibration time of 150 s was set between consecutive injections. The binding isotherms were analysed with the MicroCal Origin software. For measurements of heat of dilution, the protein

sample in the cell was replaced by Buffer C and all other conditions were kept identical.

**Measurement of cell viability.** HdhQ7 and Q111 mouse striatal cells were treated with the HV-3 peptide or the control peptide TAT (3 µM, each) in an FBS-free DMEM medium or in DMEM containing 10% serum for 24 h. Medium from the cultured cells was harvested. Proteins from the medium were purified using Amicon Ultra 0.5 ml centrifugal filters (Millipore). HMGB1 release into the medium was then analysed by Western blotting with anti-HMGB1 antibody. In parallel, cell death was determined by measuring LDH release into the culture medium, using LDH-Cytotoxicity Assay Kit II (Roche, USA) by following the manufacturer's instruction.

**Immunocytochemistry.** Cells cultured on coverslips were washed with PBS and fixed in 4% formaldehyde, and then permeabilized with 0.1% Triton X-100. After incubation with 2% normal goat serum, fixed cells were incubated overnight at 4 °C with indicated primary antibodies. Cells were washed with PBS and incubated with Alexa Fluor 568, 488 or 405 secondary antibody, followed by incubation with Hoechst dye (1:10,000; Invitrogen). Coverslips were mounted, and slides were imaged by confocal microscopy (Fluoview FV100; Olympus).

To determine mitochondrial mass in cultures, cells were stained with antibodies against Tom20 or stained with Mitotracker green. The fluorescent density of Tom20 (1:500) or mitotracker green was quantitated using NIH Image J software. To measure the membrane potential of mitochondria in cultures, cells were incubated with 0.25 µM tetra-methyl rhodamine (Invitrogen Life Science) for 20 min at 37 °C. To measure lysosomal activity, cells were incubated with Lyso-ID Red dye (Enzo Life Science) for 30 min at 37 °C. The images were visualized by microscope and quantitation of the density of red fluorescence was carried out using NIH ImageJ software. For immunocytochemistry study, at least 100 cells per group were counted and quantitated by an observer blind to experimental conditions.

In patient-iPS cell-derived neurons, to ensure the observation of mitochondria in the medium spiny neurons, the cells were stained with a mitochondrial marker (anti-TOM20, 1:500) and markers for medium spiny neurons (DARPP-32, 1:200, Epitomics). The quantitation of neurite length was conducted only in the neurons immunopositive for both DARPP-32 and Tuj1 (neuron-specific class III beta-tubulin, 1:5,000, Covance). At least 50 neurons/group were counted.

**Electron microscopy.** The HdhQ7 and HdhQ111 cells were fixed by 2.5% glutaraldehyde in 0.1 M cacodylate buffer. Small pieces of the striata tissue of mice were fixed by immersion in triple aldehyde-DMSO. After rinsing in 0.1 M phosphate buffer (pH 7.3), the samples were post-fixed in ferrocyanide-reduced osmium tetroxide.

For immunogold EM analysis of VCP, mitochondria were isolated from HdhQ7 and HdhQ111 cells. The mitochondrial pellets were fixed in 4% paraformaldehyde at 4 °C for 20 min. After washing, the pellet was blocked with 5% of normal goat serum for an hour and incubated with anti-VCP antibody (1:100) overnight at 4 °C, followed by incubation with Gold conjugate 2nd antibody (10 nm gold-conjugated goat anti-mouse IgG, British BioCell International, Ted Pella, Inc., Redding, CA). The mitochondria pellets were fixed by 2.5% glutaraldehyde in 0.1 M phosphate buffer to stabilize the gold particles. After rinsing in 0.1 M phosphate buffer (pH 7.3), they were post-fixed in ferrocyanide-reduced osmium tetroxide and embedded in 1.5% low gel temperature agarose.

The above fixed samples were rinsed by water followed by overnight soaking in acidified uranyl acetate. After rinsing in distilled water, the blocks were dehydrated in ascending concentrations of ethanol, passed through propylene oxide, and embedded in Poly/Bed resin. Thin sections were sequentially stained with acidified uranyl acetate followed by a modification of Sato's triple lead stain. These sections were examined in a FEI Tecnai Spirit (T12) transmission electron microscope with a Gatan US4000 4kx4k CCD at the Case Western Reserve University EM core facility. Mitochondria from 15 random areas in each group were imaged by an experimenter blind to the experimental groups. The length of mitochondria was measured by NIH Image J software. The number of mitophagosomes per 100 µm$^2$ was counted. The number of gold particles labelling VCP on mitochondria were quantitated.

**Animal model of HD.** All experiments in animals were conducted in accordance with protocols approved by the Institutional Animal Care and Use Committee of Case Western Reserve University and were performed based on the National Institutes of Health Guide for the Care and Use of Laboratory Animals. Sufficient procedures were employed for reducing pain or discomfort of subjects during the experiments.

Male R6/2 mice and their wt littermates (4 weeks old) were purchased from Jackson Laboratories (Bar Harbor, ME; B6CBA-TgN (HD exon 1)62; JAX stock number: 006494). These mice (C57BL/6 and CBA genetic background) are transgenic for the 5′ end of the human *HD* gene carrying 100–150 glutamine (CAG) repeats.

YAC128 (FVB-Tg(YAC128)53Hay/J, JAX stock number: 004938) breeders (FVB/N genetic background) were purchased from Jackson Laboratories. The

YAC128 mice contain a full-length human huntingtin gene modified with a 128 CAG repeat expansion in exon 1. The mice were mated, bred and genotyped in the animal facility of Case Western Reserve University. Male mice at the ages of 2, 3, 6, 9 and 12 months were used in the study.

All of the mice were maintained with a 12-h light/dark cycle (on 06:00 hours, off 18:00 hours).

**Systemic peptide treatment in HD mice.** All randomization and peptide treatments were prepared by an experimenter not associated with behavioural and neuropathology analysis.

Male hemizygous R6/2 mice (Tg) and their age-matched wt littermates (5-week-old) were implanted with a 28-day osmotic pump (Alzet, Cupertino CA) containing TAT control peptide or HV-3 peptide, which delivered peptides to the mice at a rate of 3 mg kg$^{-1}$ per day. The first pump was implanted subcutaneously in the back of 5-week-old mice between the shoulders and replaced once, after 4 weeks.

YAC128 mice (Tg) and their age-matched wt littermates were implanted with an osmotic pump containing TAT control peptide or HV-3 peptide (3 mg/kg/day, each) starting from the age of 3 months. The pump was replaced once every month. By the age of 12 months, the treatments were terminated and the mouse samples were harvested for analysis.

**Behavioural analysis in HD mice.** All behavioural analyses were conducted by an experimenter who was blind to genotypes and treatment groups.

Gross locomotor activity was assessed in R6/2 mice and age-matched wt littermates at the ages of 13 weeks and in YAC128 mice and age-matched wt littermates at the ages of 2, 3, 6, 9, and 12 months. In an activity chamber (Omnitech Electronics, Inc), mice were placed in the center of the chamber and allowed to explore while being tracked by an automated beam system (Vertax, Omnitech Electronics Inc). Distance moved, horizontal, vertical, and rearing activities were recorded. Because R6/2 mice were sensitive to changes in environment and handling, we only conducted one-hour locomotor activity analysis for R6/2 mice and wt littermates. We performed 24 h of locomotor activity analysis for YAC128 mice and their wt littermates.

Hindlimb clasping was assessed with the tail suspension test once a week from the ages 8 to 11 weeks in R6/2 mice. Mice were suspended by the tail for 60 s and the latency for the hindlimbs or all four paws to clasp was recorded using the score system[67]: Clasping over 10 s, score 3; 5–10 s, score 2; 0–5 s, score 1; 0 s, score 0.

The motor coordination and balance of YAC128 mice were tested on an accelerating Rotarod (IITC Life Sciences, Serials 8) at the ages of 2, 3, 6, 9 and 12 months. Training and baseline testing for motor function tasks were conducted at 2 months of age. For training, mice were given three 120-s trials per day at a fixed-speed of 15 r.p.m. for three consecutive days. During the testing phase, the Rotarod accelerated from 5 to 40 r.p.m. over 3 min; the maximum score was 300 s. Rotarod scores were the average of three trials per day (with 2 h rest between trials) for 3 consecutive days.

The body weight and survival rate of HD mice and wt littermates were recorded throughout the study period.

**Immunohistochemistry and stereological measurements.** Mice were deeply anaesthetized and transcardially perfused with 4% paraformaldehyde in PBS. Brains were processed for paraffin embedment. Brain sections (5 μm, coronal) were used for immunohistochemical localization of DARPP-32 (1:500, Epitomics) using the IHC Select HRP/DAB kit (Millipore). Quantitation of DARPP-32 immunostaining was conducted using NIH image J software. The same image exposure times and threshold settings were used for sections from all treatment groups.

To measure the number of NeuN-positive cells, a series of 25 μm thick coronal sections spaced 200 μm apart spanning the striatum were stained with NeuN antibody (Millipore, 1:500) and visualized by diaminobenzidine. For neuropathological analyses, brain sections were analysed stereologically. Briefly, unbiased stereological counts of NeuN-positive neurons within the striatum were performed using unbiased stereological principles and analysed with StereoInvestigator software (Microbrightfield, Williston, VT). Optical fractionator sampling was carried out on a Leica DM5000B microscope (Leica Microsystems, Bannockburn, IL) equipped with a motorized stage and Lucivid attachment (× 40 objective). The following parameters were used in the final study: grid size, (X) 500 μm, (Y) 500 μm; Counting frame, (X) 68.2 μm, (Y) 75 μm, depth was 20 μm. Gundersen coefficients of error for $m = 1$ were all less than 0.10. Stereologic estimations were performed with the same parameters in striatum of wt or YAC transgenic mice treated with the control peptide or peptide HV-3 ($n = 6$ mice per group). The total volume of stratial tissue measured in each brain is calculated by StereoInvestigator and the neuronal density is presented as NeuN positive cell number per mm$^3$.

Quantitation was conducted by an experimenter blind to the experimental groups.

**Western blot analysis.** Protein concentrations were determined by Bradford assay. Protein was resuspended in Laemmli buffer, loaded on SDS–PAGE, and transferred onto nitrocellulose membranes. Membranes were probed with the indicated antibodies, followed by visualization with ECL. Representative blots have been cropped for presentation. Images of full-size blots are presented in Supplementary Fig. 9.

**GST pull-down assay.** Bacteria-expressed GST or GST–VCP, or GST–VCP truncated mutants were immobilized on glutathione-Sepharose 4B beads (GE Healthcare) for three hours and then washed three times. Beads were incubated with total lysates of mouse brains overnight at 4 °C. Beads were then washed with a GST binding buffer (100 mM NaCl, 50 mM NaF, 2 mM EDTA, 1% Triton-X-100 and protease inhibitor cocktail) and were analysed by SDS–PAGE and IB.

**Molecular docking.** We constructed models of VCP with the homology modelling software MODELLER9.9 using the crystal structure of p97 (PDB ID 3CF1) as the template. Sequence alignment of VCP with 3CF1A (p97) showed two gaps in the 1–21 and 707–806 amino acid regions of VCP. Thus, a homologous model of the 22–706 amino acids of VCP could be readily generated. Therefore, the 1–21 and 707–806 amino acids of VCP were not taken into account in the present work. The model of HV-3 peptide was built with Amber12 in our laboratory.

All simulations were performed using the Amber12 software package together with the f99SB parameters for proteins, and the Ptraj module of Amber12 was used to analyse the computational results. The starting models were solvated in a rectangular box of TIP3P (explicit water model) water molecules with a minimum distance of 12 Å between any protein atom and the box boundaries. To neutralize the models, three chloride ions were added. Before MD simulation, a series of minimizations were performed. All water molecules were first minimized while restraining the positions of the atoms of the protein with a harmonic potential. The whole system was then energy minimized without restraint for 2,000 steps using a combination of the steepest descent and conjugated gradient methods. After gradually heating the system from 10 to 310 K over 100 ps using the NVT ensemble, a 1 ns simulation was performed at 1 atm and 300 K with the NPT ensemble to equilibrate the whole system. For production runs, MD simulations were performed in the NPT ensemble for 40 ns for VCP and for 400 ns for HV-3.

For all simulations, all bonds involving hydrogen atoms were constrained using the SHAKE algorithm. A time period of 2 fs and a non-bonded interaction cutoff radius of 10 Å were used. The particle-mesh Ewald method was employed to calculate long-range electrostatic interactions. During the sampling process, the coordinates were saved every 5 ps for further analysis.

We analysed the HV-3 and VCP with the rigid-body docking programme in Discovery Studio 2.5. The angular step size sets as 15, RMSD cutoff 6.0, interface cutoff 9.0 and 2,000 configurations are generated. Supplementary Fig. 4e showed the most plausible configuration according to the comprehensive results of maximum score in ZDOCK, Density maximum and Cluster minimum.

**Statistical analysis.** Sample sizes are determined by power analysis based on pilot data collected by our labs or published studies. In animal studies, we used $n = 15$–20 mice/group for behavioural tests, $n = 6$ mice/group for biochemical analysis and $n = 6$ mice/group for pathology studies. In cell culture studies, we performed each study with at least three independent replications. For all of the animal studies, we have ensured randomization and blinded conduct of experiments. For all imaging analysis, the quantitation was conducted by an observer who was blind to the experimental groups. No samples or animals were excluded from the analysis.

Data were analysed by Student's $t$-test or analysis of variance (ANOVA) with *post hoc* Holm-Sidak test for comparison between two groups. Survival, behavioural test and body weight were analysed by repeated-measures two-way ANOVA. Data are expressed as mean ± s.e.m. Statistical significance was considered achieved when the value of $P$ was $< 0.05$.

**Data availability.** The data that support the findings of this study are available from the corresponding author on request.

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

## Acknowledgements

The work is supported by the National Institutes of Health grant (NIH R01 NS088192) (to X.Q.), NIH R01 NS091213 (to Y.L.), NIH R01 HL052141 (to D.M.-R.), NIH R01GM108921 and the American Heart Association Scientist Development Grant 12SDG12070069 (to S.C.). We thank Dr Eliezer Masliah of the University of California at San Diego for providing the frozen postmortem brain samples of HD patients and normal subjects, thank for NIH NeuroBiobank for providing formalin-fixed brain samples of HD patients and normal subjects. We thank Dr Yongbo Song of Shenyang Pharmaceutical University of China for conducting the molecular docking analysis of VCP and HV-3.

## Author contributions

X.G. performed all experiments in cell cultures and biochemical analyses of animal models and patient samples; X.Y.S. maintained HD mice and conducted animal behavioural analysis; D.H. examined VCP in Parkinson's disease models; Y.-J.W. performed proteomic analysis of mtHtt interactors; H.F. conducted electron microscopy analysis; R.V. and S.C. purified VCP protein and performed the ITC analysis of HV-3; A.U.J. and D.M.-R. conducted the toxicity analysis of HV-3 in mice; Y.L. conducted stereology analyses of the striatal neuronal number in mice; X.Q. conceived, designed and supervised all the studies and wrote the manuscript.

## Additional information

**Competing financial interests:** A patent on the design and applications of the HV-3 peptide inhibitor has been filed. The authors declare no conflict of interest.

