## [Peer Review File · Nature Communications]

C Putative protein candidates that bind to mtHtt in HD mouse striatal cells

Category	Uniprot ID	Protein name	Symbol	Spectral counts
Protein quality control	Q01853	Valosin-containing protein	Vcp	45
	P80317	T-complex protein 1 subunit zeta	Cct6a	22
	P08113	Heat-shock protein 90	Hsp90	22
	O08583-1	THO complex subunit 4	Alyref	9
DNA/RNA binding protein	Q9CY58-3	Plasminogen activator inhibitor 1 RNA-binding protein	Serbp1	16
	A2BGG7	Nuclease-sensitive element-binding protein 1	Ybx1	9
	P60824	Cold-inducible RNA-binding protein	Cirbp	5
Mitochondrial protein	P08249	Malate dehydrogenase 2	Mdh2	13
Neurogenesis	Q6P5H2-2	Nestin	Nes	22

Supplementary Figure 1: Proteomic analysis of Htt interactors on mitochondria in HD knock-in mouse striatal cells. (a) Mitochondria were isolated from either HdhQ7 or HdhQ111 mouse striatal cells and subjected to immunoprecipitation (IP) with anti-Htt antibodies (MAB2166, recognizing full-length wtHtt or mtHtt). The immunoprecipitates were eluted from beads followed by tandem mass spectrometry analysis. (b) Venn diagram showing the protein numbers and overlaps of Htt binding proteins on the mitochondria in HdhQ7 and HdhQ111 cells. Four proteins were identified in both HdhQ7 and Q111 mitochondria, nine proteins only in mitochondria of HdhQ111 cells, and four proteins only in mitochondria of HdhQ7 cells. (c) Putative protein candidates that bind to mtHtt on mitochondria of HD HdhQ111 cells. Spectral counts were used here as semi-quantitation (1).

Supplementary Figure 2: VCP localization on subcellular organelles and the information of HD patient samples. (a) Localization of VCP (red) in the ER (green, marked with anti-calnexin antibody) or endosome (green, marked with anti-EEA1 antibody) was determined by confocal microscopy. Pearson's coefficient was used to quantitate the extent of colocalization of VCP/calnexin or VCP/EEA1. Nuclei were stained with DAPI. The data are mean \pm SEM of three independent experiments. Scale bar is 10 μ m. (b) HEK293 cells were transfected with Parkinson's disease-associated mutants LRRK2 G2019S or alpha-synuclein A53T for 24 hours. The control vector, wildtype LRRK2 or wildtype alpha-synuclein, was also expressed for comparison. Mitochondria were isolated at the indicated groups and VCP mitochondrial levels were examined by Western blot analysis. There was no evidence of VCP translocation to mitochondria observed in these mutants-expressing cells relative to those in cells expressing wild-type counterparts. The representative blots were from two to three independent experiments. (c) Data of normal subjects and HD patients shown in Fig. 2D. The evaluation of neuropathology of HD patients was summarized. (d) Total cortical protein lysates from postmortem brain tissues of a normal subject (5248) and a HD patient (5374) were subjected to IP with anti-VCP antibody followed by Western blot analysis with anti-1C2 antibody. Arrows indicate mtHtt recognized by 1C2 antibody which does not detect wt Htt. There was no binding between VCP and mtHtt observed in this patient brain tissue. The HD patient (5374) had very subtle loss of neurons based on the neuropathology evaluation shown in (c).

HD mouse striatal cells

Supplementary Figure 3: Htt and mtHtt are not substrates of VCP. Left: HdhQ7 and HdhQ111 cells were treated with MG132 (10 μM for 16 hours). Protein levels of Htt were determined in the total lysates of the cells. Actin was used as a loading control. The representative blots were from three independent experiments. Right: HdhQ7 and HdhQ111 were treated with Eer I (Eeyarestatin I, 5 μM for 16 hours). Protein levels of Htt were determined. Mcl-1, a mitochondrial Bcl-2 family protein, was used as a positive control. Mcl-1 has been reported to be a substrate of VCP on mitochondria and it is degraded via the ubiquitin-proteasome system. The representative blots were from three independent experiments.

Supplementary Figure 4: Rational design of a peptide inhibitor to block VCP/Htt interaction. (a) GST or GST-VCP was incubated with total lysates of mouse brain and the indicated peptides (3 μ M) for 16 hours, followed by immunoblotting with anti-Htt antibodies. Representative blots are from three independent experiments. Quantitation of VCP/Htt binding was shown below the blot as the mean of three independent experiments. Data are mean \pm SEM. *, $p < 0.05$ vs. TAT-treated group (Paired Student's *t* test). (b) HEK293 cells were co-expressed with GFP-VCP and Myc-73Q FL plasmids as indicated. After 48 hours incubation with the indicated peptides (3 μ M/day, each), immunoprecipitation analysis was performed. The shown blots are from three independent experiments. (c) HEK293 cells were co-expressed with GFP-VCP and Myc-73Q FL plasmids as indicated. After 48 hours incubation with HV-3 at various dosages arranging from 0 to 4 μ M, immunoprecipitation analysis was performed followed by western blot analysis. The blots of the bindings of mtHtt/VCP were quantitated with NIH Image J software. The IC₅₀ was calculated. (d) Upper: sequence of the HV-3 peptide and the control peptide TAT. Lower: HV-3 peptide sequence is highly conserved among species. (e) The HV-3 peptide was docked to the VCP. Upper: Cartoon representing the predicted VCP structure using the mouse crystal structure of p97 (PDB ID 3CF1); sticks represent the structure of HV-3 (see supplementary method). The blue is the N-terminal and the red is the C-terminal of VCP. The HV-3 binds to the hydrophilic cavity of the VCP surface (marked in

green). Lower: the enlarged area is labeled in green. (f) The sequence in VCP corresponding to HV-3 in Htt was deleted (Δ VCP). GST, GST-VCP, or GST- Δ VCP was incubated with total lysates of mouse brain for 16 hours followed by Western blot with anti-Htt antibodies. The representative blots are from three independent experiments.

Supplementary Figure 5: Treatment of peptide inhibitor HV-3 in HD animal models. (a) HD R6/2 mice and wildtype littermates were treated with either a control peptide or peptide HV-3 (3 mg/kg/day, subcutaneous administration with an Alzet osmotic pump) from 5 to 13 weeks of age. The pump was replaced every four weeks. YAC128 mice and wildtype littermates were treated with control peptide TAT or HV-3 peptides (3 mg/kg/day, each, subcutaneous administration with an Alzet osmotic pump) from the age of 3 months to the age of 12 months. Mouse behaviors and HD-associated pathology were determined every three months after beginning treatment. The pump was replaced every four weeks. (b) Total protein lysates were extracted from HdhQ111 striatal cells and the striata of HD R6/2 and YAC128 mice. VCP protein levels were determined by Western blot analysis. Actin was used as a loading control. Representative blots from three independent experiments are shown.

Supplementary Figure 6: Peptide HV-3 does not affect ER stress response and apoptosis. (a) HD mouse striatal cells were treated with thapsigargin (1µM) or tunicamycin (10µg/ml) for 16 hours in the presence or absence of HV-3 (3 µM). GRP78 total protein levels were determined by western blot analysis. Representative blots of three independent experiments are shown. Actin was used here as a loading control. (b) Twenty-four hours of serum starvation was performed in HD mouse striatal cells. Caspase-3 activity analysis was performed in the indicated groups. Data are mean ± SEM of three independent experiments. (c) Flag-VCP and GFP-LC3B were co-transfected into HdhQ7 striatal cells. Total lysates were isolated after 36 hours of transfection. The GFP-LC3B levels were examined by western blot analysis. Actin was used as a loading control. Histogram: quantitation of GFP-LC3B total protein level. Data are mean ± SEM of three independent experiments.

Supplementary Figure 7: VCP accumulation on mitochondria elicits excessive mitophagy. (a) HeLa cells were transfected with either the control vector or flag-mtVCP for 36 hours. Cells were stained with flag (red) and Tom20 (green) antibodies. Mitochondrial morphology was analyzed. Scale bar: 10 μ m. The appearance of mitochondrial aggregates (b), mitochondrial mass (c), and mitochondrial membrane potential (d) in cells expressing flag-mtVCP were quantitated as described in the Method section. Data are mean \pm SEM of three independent experiments. Paired Student's *t* test. (e) HeLa cells were transfected with flag-mtVCP and GFP-LC3B for 36 hours. Cells were stained with flag (blue) and Tom20 (red) antibodies. Scale bar: 10 μ m. GFP-LC3B/Tom20 colocalization was determined using confocal microscopy. At least 100 cells per group were counted by an observer blind to experimental conditions. The data are mean \pm SEM from three independent experiments. Paired Student's *t* test. (f) Wild-type striatal cells were transfected with either the control vector or flag-mtVCP for 36 hours. Mitochondria were isolated. The autophagic flux of mitochondria was determined by quantitation of LC3 II mitochondrial levels in the presence or absence of BFA (20 nM). The data are mean \pm SEM from three independent experiments. Paired Student's *t* test. (g) Parkin total protein levels were determined in HdhQ7 and HdhQ111 cells. Parkin in HEK293 cells was used as a positive control. Actin was a loading control. There was no Parkin expression in HdhQ7 and Q111 cells. The representative data are from two independent experiments.

Supplementary Figure 8: Toxicity analysis of HV-3 in naïve mice.

We performed a toxicity analysis of HV-3 in mice. Six-week-old C57Black mice were treated with saline or peptide HV-3 at 3mg/kg/day by Alzet pump implantation. After 4-week treatment, the plasma, blood and whole mouse bodies of mice were collected. Examination on blood chemistry (Supplementary Table 1), hematology (Supplementary Table 2) and general necropsy of mice were performed at the Animal Diagnostic Laboratory at Stanford University. The experimenter who conducted the analysis was blind to the treatment groups. Automated hematology was performed on the Sysmex XT-2000iV analyzer system. Blood smears were made for all full CBC samples and reviewed by a medical technologist. Manual differentials were performed as indicated by species and automated analysis. Chemistry analysis was performed on the Siemens Dimension Xpand analyzer.

(a) Frozen tissue sections of brain and spleen were stained with anti-CD3 antibody by immunohistochemistry. There is no observed difference in the immuno-density of CD3 between saline- and HV-3-treated groups. Scale bar: 100 μ m. (b) The size and weight of spleens in mice treated with saline or peptide HV-3 were compared. Left: the ratio of spleen weight to body weight was calculated and expressed as mean \pm SEM of five mice in each group. Right: the images of spleens of mice treated with saline or peptide HV-3. There is no observed difference in the size of the spleens. Paired Student's *t* test. Four-week sustained treatment with HV-3 by pump implantation had no effects on blood chemistry (Supplementary Table 1), hematology (Supplementary Table 2), and gross necropsy of naïve mice when compared to mice treated with saline. A four-week treatment with HV-3 had minor effects on immuno-density of CD3, a marker of T-cell for adaptive immune response, in the brains and spleens of naïve mice (a). Treatment with HV-3 had no effects on the size and weight of the spleens (b).

Reference:

1. Liu H, Sadygov RG, and Yates JR, 3rd. A model for random sampling and estimation of relative protein abundance in shotgun proteomics. *Anal Chem.* 2004;76(14):4193-201.

Figure1B

Figure1C

Figure1F

Figure2A left panel

Figure2A right panel

Figure2B

Figure2C

Supplementary Figure 9: Full scan of western blots. Specific bands that were cropped and used in the figures are marked in the boxes.

Figure2D

Figure3C

Figure3D upper panel

Figure3D lower panel

Figure3F left panel

Figure3F right panel

Figure3G

Figure3H

Figure3I

Figure4A

Figure4D

Supplementary Figure 9: Full scan of western blots (continued).

Specific bands that were cropped and used in the figures are marked in the boxes.

Figure5A

Figure5B

Figure6B

Figure6C

Figure8A

Suppl2B left panel

Suppl2B right panel

Suppl3

Suppl2D

Suppl4A

Supplementary Figure 9: Full scan of western blots (continued).

Specific bands that were cropped and used in the figures are marked in the boxes.

Suppl4B

Suppl4F

Suppl5

Suppl6A

Suppl6C

Suppl7G

Supplementary Figure 9: Full scan of western blots (continued).

Specific bands that were cropped and used in the figures are marked in the boxes.

Table 1: Effects of 4-week treatment with HV-3 on blood biochemistry of naïve mice

	Saline		HV-3		p-value	unit	n
	Mean	SD	Mean	SD			
Glucose	304.00	27.78	312.25	27.96	0.69	mg/dL	4/ group
AST	247.00	54.30	237.25	63.38	0.82	U/L	
ALT	63.25	20.89	71.00	19.20	0.60	U/L	
Alkaline Phosphatase	126.75	6.45	108.00	22.93	0.17	IU/L	
Total Bilirubin	0.15	0.06	0.13	0.05	0.54	mg/dL	
BUN	35.00	9.49	27.50	2.38	0.18	mg/dL	
Creatinine	0.38	0.10	0.40	0.08	0.70	mg/dL	
T.Protein	5.05	0.13	5.15	0.17	0.39	g/dL	
Albumin	2.90	0.08	2.95	0.13	0.54	g/dL	
Globulin	2.15	0.19	2.20	0.12	0.67		
Sodium	149.75	1.26	151.25	0.96	0.11	mmol/L	
Potassium	8.40	0.62	7.28	0.75	0.06	mmol/L	
Calcium	9.35	0.25	9.78	0.21	0.04	mg/dL	
Phosphorus	8.48	1.02	8.45	1.09	0.97	mg/dL	
Chloride	107.75	2.06	109.00	0.82	0.30	mmol/L	
Carbon Dioxide	17.18	1.56	16.45	2.34	0.62	mmol/L	
Na/K Ratio	17.91	1.43	20.96	2.20	0.06		
Anion Gap	33.23	3.55	33.08	2.46	0.95	mmol/L	

Table 2: Effects of 4-week treatment with HV-3 on hematology of naïve mice

	Saline		HV-3		p-value	unit	n
	Mean	SD	Mean	SD			
WBC	3.30	1.74	3.39	1.12	0.94	K/uL	4/ group
RBC	9.11	0.63	9.37	0.95	0.68	M/uL	
Hgb	13.33	0.89	13.93	1.24	0.48	gm/dL	
HCT	42.95	2.44	44.60	3.98	0.52	%	
MCV	47.20	0.88	47.67	0.71	0.49	fL	
MCH	14.65	0.24	14.90	0.20	0.20	pg	
MCHC	31.03	0.51	31.27	0.06	0.46	g/dL	
Platelet count	904.00	492.04	1021.33	199.89	0.72	K/uL	
RDW	21.25	1.14	21.77	1.12	0.58	%	
Reticulocyte Count	4.96	1.12	4.30	0.35	0.38	%	
IRF	56.85	3.16	53.97	1.63	0.21	%	
Neutrophils	853.50	742.59	1525.67	884.88	0.32		
Lymphocytes	2328.75	1036.07	1645.00	270.94	0.33		
Monocytes	80.75	24.92	182.33	95.57	0.09		
Eosinophils	39.50	37.47	37.00	46.36	0.94